# Learning from Asymmetrically-corrupted Data in Regression for Sensor Magnitude

## Abstract

This paper addresses a regression problem in which output label values represent the results of sensing the magnitude of a phenomenon. A low value of such labels can either mean that the actual magnitude of the phenomenon has been low or that the sensor has made an incomplete observation. This leads to a bias toward lower values in labels and its resultant learning because labels for incomplete observations are recorded as lower than those for typical observations, even if both have monitored similar phenomena. Moreover, because an incomplete observation does not provide any tags indicating incompleteness, we cannot eliminate or impute them. To address this issue, we propose a learning algorithm that explicitly models the incomplete observations to be corrupted with an asymmetric noise that always has a negative value. We show that our algorithm is unbiased with a regression learned from the uncorrupted data that does not involve incomplete observations. We demonstrate the advantages of our algorithm through numerical experiments.

## 1 Introduction

This paper addresses a regression problem for predicting the magnitude of a phenomenon when an observed magnitude involves a particular measurement error. The magnitude typically represents *how large a phenomenon is or how strong the nature of the phenomenon is*. Such examples of predicting the magnitude are found in several application areas, including pressure, vibration, and temperature (Vandal et al., 2017; Shi et al., 2017; Wilby et al., 2004; Tanaka et al., 2019). In medicine and healthcare, the magnitude may represent pulsation, respiration, or body movements (Inan et al., 2009; Nukaya et al., 2010; Lee et al., 2016; Alaziz et al., 2016; 2017; Carlson et al., 2018).

More specifically, we learn a regression function to predict the *label* representing the magnitude of a phenomenon from *explanatory variables*. The training data consists of pairs of the label and explanatory variables, but note that the label in the data is observed with a sensor and is not necessarily in agreement with the actual magnitude of the phenomenon. We note that we use the term "label" even though we address the regression problem, and it refers to a real-valued label in this paper.

In the example of predicting the magnitude of body movements, the label in the data is measured with an intrusive sensor attached to the chest or the wrist, and the explanatory variables are the values measured with non-intrusive bed sensors (Mullaney et al., 1980; Webster et al., 1982; Cole et al., 1992; Tryon, 2013). A regression function for this example would make it possible to replace intrusive sensors with non-intrusive ones, which in turn will reduce the burden on patients.

Although the sensors that measure the label generally have high accuracy, they often make incomplete observations, and *such incomplete observations are recorded as low values instead of missing values*. This leads to the particular challenge where a low value of the label can either mean that the actual magnitude of the phenomenon has been low or that the sensor has made an incomplete observation, and there are no clues that allow us to tell which is the case. We illustrate this challenge in Fig. 1-(a).

Such incomplete observations are prevalent in measuring the magnitude of a phenomenon. For example, the phenomenon may be outside the coverage of a sensor, or the sensing system may experience temporal mechanical failures. In the example of body movements, the sensor may be temporarily detached from the chest or wrist. In all cases, the sensor keeps recording low values, while the actual magnitude may be high, and no tag indicating incompleteness can be provided.

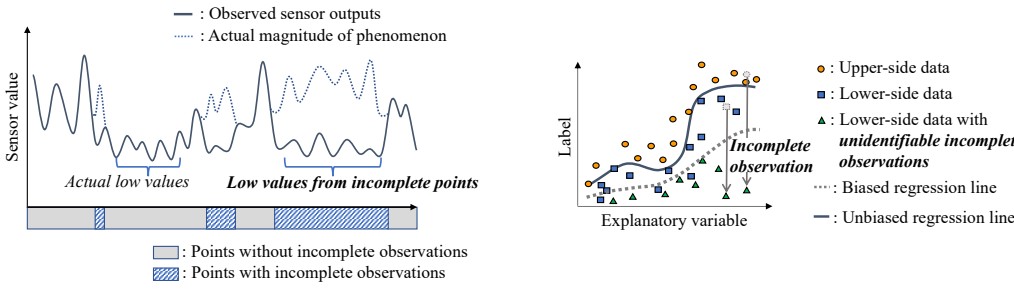

(a) Sensing results for magnitude of phenomenon      (b) Asymmetrically-corrupted data

Figure 1: (a) Low sensor value can either mean actual low magnitude or incomplete observation. (b) Labels for incomplete observations, depicted as triangles, become lower than those of typical observations, depicted as circles or squares.

This incomplete observation is particularly severe for the sensor measuring the label since it is single-source and has narrower data coverage. This stems from the fact that the sensor is usually intrusive or it is costly to produce highly accurate observations for measuring the label. Examples of this can be seen in chest or wrist sensors that focus on the movements of a local body part with high accuracy and often miss movements outside their coverage, such as those of parts located far from where the sensor is attached. At most, a single intrusive sensor can be attached to a patient to avoid burdening them. In contrast, the sensors measuring the explanatory variables are usually multi-source and provide broader data coverage. For example, multiple sensors can be attached to various places of a bed and globally monitor the movements of all body parts on the bed but with lower accuracy.

One cannot simply ignore the problem that the observations of labels may be incomplete because the estimated regression functions trained on such data with incomplete observations are severely biased toward lower values regardless of the amount of available training data. This bias comes from the fact that incomplete observations always have lower values than the actual magnitude of a phenomenon, and they occur intensively on label sensors, while explanatory variables are usually observed completely. Moreover, incomplete observations can be much more frequent than expected.

Unfortunately, since we cannot identify which observations are incomplete, we cannot eliminate or impute them by using existing methods that require identifying incomplete observations. Such methods include thresholding, missing value detection (Pearson, 2006; Qahtan et al., 2018), imputation (Enders, 2010; Smieja et al., 2018; Ma & Chen, 2019; Sportisse et al., 2020), and semi-supervised regression (Zhou & Li, 2005; Zhu & Goldberg, 2009; Jean et al., 2018; Zhou et al., 2019).

The issues of incomplete observations also cannot be solved with robust regression (Huber et al., 1964; Narula & Wellington, 1982; Draper & Smith, 1998; Wilcox, 1997), which takes into account the possibility that the observed labels contain outliers. While robust regression is an established approach and state-of-the-art against corrupted labels in regression, it assumes *symmetric label corruption*. Namely, the noise is assumed to not be biased either positively or negatively. Since incomplete observations induce the noise that is severely biased toward lower values, robust regression methods still produce regression functions that are biased toward lower values than the one that would be learned from the data without incomplete observations.

In this paper, to mitigate the bias toward lower values, we explicitly assume the existence of the noise from incomplete observations, which always has negative values, in addition to the ordinary symmetric noise. That is, we consider our training data to be *asymmetrically-corrupted data*. We then formulate a regression problem from our asymmetrically-corrupted data and design a principled learning algorithm for this regression problem.

By explicitly modeling the incomplete observation, we derive a learning algorithm that has a rather drastic feature: namely, it ignores the labels that have relatively low values (lower-side labeled data). In other words, our algorithm uses the data whose labels have relatively high values (upper-side labeled data) and the data whose labels are ignored (unlabeled data). Hence, we refer to our algorithm

as *upper and unlabeled* regression (U2 regression). This aligns with the intuition that the labels with low values are unreliable, since those low values may be due to incomplete observations.

Our main result is that U2 regression, which learns from the asymmetrically-corrupted data, produces a regression function that is, under some technical assumptions, unbiased and consistent with the one that is produced from the uncorrupted data that does not involve incomplete observations. This counterintuitive result is achieved by considering a specific class of loss functions and deriving their *gradient*, which only requires upper-side labeled data and unlabeled data in the asymmetrically-corrupted data and can still be shown to be asymptotically equivalent to the expression of the gradient that has access to the uncorrupted data. The main novelty in our approach is thus in the loss function, and we will empirically demonstrate the effectiveness of the proposed class of loss functions over existing common loss functions in dealing with asymmetrically-corrupted data in synthetic and six real-world regression tasks.

**Contributions.** The main contributions of this paper are summarized as follows.

- We formulate a novel problem of learning a regression function from asymmetrically-corrupted data. This is important for applications where the magnitude of a phenomenon is measured with a sensor that is susceptible to unidentifiable incomplete observations.

- We derive an unbiased and consistent learning algorithm (U2 regression) for this problem from the new class of loss functions.

- Extensive experiments on synthetic and six real-world regression tasks including a real use case for healthcare demonstrate the effectiveness of the proposed method.

## 2 REGRESSION FROM ASYMMETRICALLY-CORRUPTED DATA

Our goal is to derive a learning algorithm with asymmetrically-corrupted data, i.e., labels in the training data are corrupted with negative-valued noise due to incomplete observations, in a manner that is unbiased and consistent with the regression that uses uncorrupted data without involving incomplete observations. We first consider the regression problem that uses the uncorrupted data in Section 2.1 and then formulate learning from the asymmetrically-corrupted data in Section 2.2.

### 2.1 REGRESSION PROBLEM FROM DATA WITHOUT INCOMPLETE OBSERVATIONS

Let $\boldsymbol{x} \in \mathbb{R}^D (D \in \mathbb{N})$ be a $D$-dimensional explanatory variable and $y \in \mathbb{R}$ be a real-valued label. We assume that, without incomplete observations, $y$ is observed in accordance with

$$y = f^*(\boldsymbol{x}) + \epsilon_{\mathrm{s}}, \tag{1}$$

where $f^*$ is the oracle regressor and $\epsilon_{\mathrm{s}}$ is the symmetric noise with 0 as the center, such as additive white Gaussian noise (AWGN).

We learn a regression function $f(\boldsymbol{x})$ that computes the value of the estimation of a label, $\hat{y}$, for a newly observed $\boldsymbol{x}$ as $\hat{y} = f(\boldsymbol{x})$. The optimal regression function, $\hat{f}$, is given by

$$\hat{f} \equiv \arg\min_{f \in \mathcal{F}} \mathcal{L}(f), \tag{2}$$

where $\mathcal{F}$ is a hypothesis space for $f$, and $\mathcal{L}(f)$ is the expected loss when the regression function $f(\boldsymbol{x})$ is applied to data $(\boldsymbol{x}, y)$, distributed in accordance with an underlying distribution $p(\boldsymbol{x}, y)$:

$$\mathcal{L}(f) \equiv \mathbb{E}_{p(\boldsymbol{x},y)}[L(f(\boldsymbol{x}), y)], \tag{3}$$

where $\mathbb{E}_p[\bullet]$ denotes the expectation over the distribution $p$, and $L(f(\boldsymbol{x}), y)$ is the loss function between $f(\boldsymbol{x})$ and $y$, e.g., the squared loss, $L(f(\boldsymbol{x}), y) = \|f(\boldsymbol{x}) - y\|^2$. The expectation $\mathbb{E}_{p(\boldsymbol{x},y)}$ can be estimated by computing a sample average for the training data $\mathcal{D} \equiv \{(\boldsymbol{x}_n, y_n)\}_{n=1}^N$, which is $N$ pairs of explanatory variables and labels.

## 2.2 Regression Problem from Asymmetrically-corrupted Data

In this paper, we consider a scenario in which we only have access to the asymmetrically-corrupted data $\mathcal{D}' \equiv \{(\boldsymbol{x}_n, y_n')\}_{n=1}^N$, where a label $y'$ may be corrupted due to incomplete observations. A corrupted label $y'$ is observed from the uncorrupted $y$ with an asymmetric negative-valued noise, $\epsilon_a$:

$$y' = y + \epsilon_a, \tag{4}$$

where the asymmetric noise $\epsilon_a$ always has a random negative value, which means $y' \leq y$.

Using only $\mathcal{D}'$, we learn a regression function $f(\boldsymbol{x})$ *as the solution for Equation 2 in an unbiased and consistent manner*. Although AWGN can be handled even when we use a naive regression method such as least squares, the asymmetric noise $\epsilon_a$, which always has a negative value, is problematic.

Intuitively, the asymmetric noise $\epsilon_a$ makes *lower-side labeled data* particularly unreliable and inappropriate for learning, while keeping *upper-side labeled data* reliable, where the upper-side labeled data refers to the data $\{(\boldsymbol{x}, y)\}$ whose label is above the regression line (i.e., $f(\boldsymbol{x}) \leq y$) and the lower-side labeled data refers to the data whose label is below the regression line. The regression line represents the estimation of a regression function. Figure 1-(b) illustrates this as a scatter plot of the value of the label against the value of an explanatory variable. Here, the data with incomplete observations appear only in the lower side of the regression line because $\epsilon_a$ makes observations have lower label values than those of typical observations, where the regression line represents such typical observations. This asymmetry leads to biased learning compared with the learning from the uncorrupted data without incomplete observations.

To address the asymmetric noise $\epsilon_a$ and its resultant bias, we formalize the assumption on the observation process for the asymmetrically-corrupted data $\mathcal{D}'$ and derive a lemma representing the nature of $\mathcal{D}'$. Then, we propose a learning algorithm based on the lemma in the next section.

The observation processes of $\mathcal{D}$ and $\mathcal{D}'$ are formally characterized as follows.

**Assumption 2.1.** *Assume $\epsilon_s \perp\!\!\!\perp f^*(\boldsymbol{x})$, $\mathbb{E}_{p(\epsilon_s)}[\epsilon_s] = 0$; $\epsilon_a \perp\!\!\!\perp f^*(\boldsymbol{x})$, $\epsilon_a \leq 0$ almost surely (a.s.); $2|\epsilon_s| < |\epsilon_a|$ a.s. when $\epsilon_a < 0$; and $\{(\boldsymbol{x}_n, y, y_n')\}_{n=1}^N$ are i.i.d. observations in accordance with Equation 1 and Equation 4,*

This assumption means that $\mathcal{D}'$ has enough information to estimate $f$, and the asymmetric noise $\epsilon_a$ is significant enough compared to the symmetric noise $\epsilon_s$, which are necessary assumptions so that the learning problem is solvable, and $\epsilon_a$ should be handled separately from $\epsilon_s$.

From Assumption 2.1, we then have the following lemma.

**Lemma 2.2.** *Let $\mathcal{F}' \equiv \{f \in \mathcal{F} : |f(\boldsymbol{x}) - f^*(\boldsymbol{x})| \leq |\epsilon_s| \ a.s.\}$. When $f \in \mathcal{F}'$, the following holds for $y \equiv f^*(\boldsymbol{x}) + \epsilon_s$ and $y' \equiv y + \epsilon_a$ under Assumption 2.1:*

$$\mathbb{E}_{p(\boldsymbol{x}, y'|f(\boldsymbol{x}) \leq y')}[G(\boldsymbol{x}, y')] = \mathbb{E}_{p(\boldsymbol{x}, y|f(\boldsymbol{x}) \leq y)}[G(\boldsymbol{x}, y)] \tag{5}$$

*for any function $G : \mathbb{R}^D \times \mathbb{R} \to \mathbb{R}$ as long as the expectations exist.*

*Proof.* We outline a proof here and provide a complete one in Appendix A.1. We first show that $\epsilon_a$ does not change the distribution for upper-side labeled data ($f^*(\boldsymbol{x}) \leq y'$) on the basis of the oracle regression function $f^*$ before and after adding $\epsilon_a$, i.e., $\epsilon_a = 0$ when $f^*(\boldsymbol{x}) \leq y'$. With the condition $f \in \mathcal{F}'$, we can further prove that $\epsilon_a = 0$ when $f(\boldsymbol{x}) \leq y'$, which is for upper-side labeled data on the basis of $f$. This establishes $p(x, y'|f(\boldsymbol{x}) \leq y') = p(x, y|f(\boldsymbol{x}) \leq y)$ and implies Lemma 2.2. □

The condition parts of these conditional distributions represent the relationships between labels and the estimations of the regression function $f$, e.g., $p(\boldsymbol{x}, y|f(\boldsymbol{x}) \leq y)$ is the distribution of $x$ and $y$ when $y$ is higher than what is given by $f$. The condition $f \in \mathcal{F}'$ represents our natural expectation that the regression function $f$ well approximates $f^*$.

Lemma 2.2 shows that $\epsilon_a$ does not change the expectation for our upper-side labeled data ($f(\boldsymbol{x}) \leq y'$) before and after adding $\epsilon_a$, which makes them still reliable for regression. In the next section, we derive an unbiased learning algorithm based on this lemma.

## 3    U2 REGRESSION

We seek to find the minimizer of the objective in Equation 2 *from the asymmetrically-corrupted data* $\mathcal{D}'$. To this end, we propose a gradient that relies only on the knowledge of the distribution of the corrupted data $p(\boldsymbol{x}, y')$ but is still equivalent to the gradient of Equation 3, which relies on the knowledge of the distribution of the uncorrupted data $p(\boldsymbol{x}, y)$. Based on Lemma 2.2, we rewrite the gradient based on $p(\boldsymbol{x}, y)$ into the one that only requires $p(\boldsymbol{x}, y')$.

### 3.1    GRADIENT FOR LEARNING FROM ASYMMETRICALLY-CORRUPTED DATA

Here, we address Equation 2 with the gradient descent. At step $t + 1$ in the gradient descent, the gradient of Equation 3 with respect to the parameters $\boldsymbol{\theta}$ of $f$ is represented with a regression function, $f_t$, which is estimated at step $t$, as follows:

$$\nabla \mathcal{L}(f_t) \equiv \mathbb{E}_{p(\boldsymbol{x},y)}[\nabla L(f_t(\boldsymbol{x}), y)], \qquad \text{where } \nabla L(f_t(\boldsymbol{x}), y) \equiv \frac{\partial L(f(\boldsymbol{x}), y)}{\partial \boldsymbol{\theta}}\bigg|_{f=f_t}. \qquad (6)$$

Note that this holds for any step in the gradient descent. When $t = 0$, $f_0$ is the initial value of $f$, and when $t = \infty$, we suppose $f_\infty = \hat{f}$. We can decompose $\nabla \mathcal{L}(f_t)$ as

$$\begin{aligned}\nabla \mathcal{L}(f_t) =& p(f_t(\boldsymbol{x}) \leq y)\mathbb{E}_{p(\boldsymbol{x},y|f_t(\boldsymbol{x})\leq y)}[\nabla L(f_t(\boldsymbol{x}), y)] \\ &+ p(y < f_t(\boldsymbol{x}))\mathbb{E}_{p(\boldsymbol{x},y|y<f_t(\boldsymbol{x}))}[\nabla L(f_t(\boldsymbol{x}), y)].\end{aligned} \qquad (7)$$

We then assume that, when $y < f(\boldsymbol{x})$, the gradient of the loss function does not depend on $y$ and only depends on $f(\boldsymbol{x})$; thus we write $\nabla L(f(\boldsymbol{x}), y)$ as $\boldsymbol{g}(f(\boldsymbol{x}))$ when $y < f(\boldsymbol{x})$ to emphasize this independence. Formally,

**Condition 3.1.** *Let* $\boldsymbol{g}(f(\boldsymbol{x}))$ *be* $\nabla L(f(\boldsymbol{x}), y)$ *for* $y < f(\boldsymbol{x})$. $\boldsymbol{g}(f(\boldsymbol{x}))$ *is a gradient function depending only on* $f(\boldsymbol{x})$ *and not on the value of* $y$.

Such common losses are the absolute loss and pinball loss, which are respectively used in least absolute regression and quantile regression and work well on real data (Lee et al., 2016; Yeung et al., 2002; Wang et al., 2005; Srinivas et al., 2020). For example, the gradient of the absolute loss is

$$\frac{\partial |f(\boldsymbol{x}) - y|}{\partial \boldsymbol{\theta}} = \frac{\partial f(\boldsymbol{x})}{\partial \boldsymbol{\theta}} \qquad \text{when } y < f(\boldsymbol{x}), \qquad (8)$$

which does not depend on the value of $y$ but only on $f(\boldsymbol{x})$.

We now propose a gradient that does not rely on the knowledge of $p(\boldsymbol{x}, y)$ but instead uses only $p(\boldsymbol{x}, y')$. Namely,

$$\begin{aligned}\nabla \tilde{\mathcal{L}}(f_t) \equiv& p(f_t(\boldsymbol{x}) \leq y)\mathbb{E}_{p(\boldsymbol{x},y'|f_t(\boldsymbol{x})\leq y')}\Big[\nabla L(f_t(\boldsymbol{x}), y)\Big] \\ &+ \mathbb{E}_{p(\boldsymbol{x})}\Big[\boldsymbol{g}(f_t(\boldsymbol{x}))\Big] - p(f_t(\boldsymbol{x}) \leq y)\mathbb{E}_{p(\boldsymbol{x}|f_t(\boldsymbol{x})\leq y')}\Big[\boldsymbol{g}(f_t(\boldsymbol{x}))\Big].\end{aligned} \qquad (9)$$

In Section 3.2, we will formally establish the equivalence between the gradient in Equation 9 and that in Equation 6 under our assumptions. Note that in the second and third terms of Equation 9, we apply expectations over $p(\boldsymbol{x})$ and $p(\boldsymbol{x}|f_t(\boldsymbol{x}) \leq y')$ to $\boldsymbol{g}(f(\boldsymbol{x}))$, even though $\boldsymbol{g}(f(\boldsymbol{x}))$ is defined to be the gradient $\nabla L(f(\boldsymbol{x}), y)$ for $y < f(\boldsymbol{x})$. This is tractable due to the nature of $\boldsymbol{g}(f(\boldsymbol{x}))$, which only depends on $f(\boldsymbol{x})$ and does not depend on the value of $y$.

Since the expectations in Equation 9 only depend on $\boldsymbol{x}$ and $y'$, they can be estimated by computing a sample average for our asymmetrically-corrupted data $\mathcal{D}'$ as

$$\nabla \hat{\mathcal{L}}(f_t) = \frac{\pi_{\text{up}}}{n_{\text{up}}}\Bigg[\sum_{(\boldsymbol{x},y)\in\{\boldsymbol{X}_{\text{up}}, \boldsymbol{y}'_{\text{up}}\}} \nabla L(f_t(\boldsymbol{x}), y)\Bigg] + \frac{1}{N}\Bigg[\sum_{\boldsymbol{x}\in\boldsymbol{X}_{\text{un}}} \boldsymbol{g}(f_t(\boldsymbol{x}))\Bigg] - \frac{\pi_{\text{up}}}{n_{\text{up}}}\Bigg[\sum_{\boldsymbol{x}\in\boldsymbol{X}_{\text{up}}} \boldsymbol{g}(f_t(\boldsymbol{x}))\Bigg],$$

$$(10)$$

where $\{\boldsymbol{X}_{\text{up}}, \boldsymbol{y}'_{\text{up}}\}$ represents the set of coupled pairs of $\boldsymbol{x}$ and $y'$ in the upper-side labeled sample set, $\{x, y' : f_t(\boldsymbol{x}) \leq y'\}$, in $\mathcal{D}'$; $\boldsymbol{X}_{\text{un}}$ is a sample set of $\boldsymbol{x}$ in $\mathcal{D}'$ ignoring labels $y'$; $n_{\text{up}}$ is the number of samples in the upper-side labeled set; $\pi_{\text{up}}$ is $\pi_{\text{up}} \equiv p(f_t(\boldsymbol{x}) \leq y)$.

Note that $\pi_{\mathrm{up}}$ depends on *the current estimation of the function $f_t$ and the label $y$ with complete observation*. Thus, it changes at each step of the gradient descent, and we cannot determine its value in a general way. In this paper, we propose a simple approach of choosing $\pi_{\mathrm{up}}$ as a single value of the hyperparameter. We optimize it with the grid search based on the validation set, which enables us to flexibly handle data variation. We will show that it works practically in our experiments.

As we will show in Section 3.2, we can use Equation 10 to design an algorithm that gives an unbiased and consistent regression function. By using the gradient in Equation 10, we can optimize Equation 2 and learn the regression function only with upper-side labeled samples and unlabeled samples from $\mathcal{D}'$ independent of lower-side labels. This addresses the issue that our lower-side labeled data is particularly unreliable and leads to overcoming the bias that stems from this unreliable part of the data. We refer to our algorithm as *upper and unlabeled* regression (U2 regression).

See Appendix B for the specific implementation of the algorithm based on stochastic optimization. The gradient in Equation 10 can be interpreted in an intuitive manner. The first term in Equation 10 has the effect of minimizing the upper-side loss. Recall that the upper-side data are not affected by the asymmetric noise under our assumptions. Thus, U2 regression seeks to learn the regression function $f$ on the basis of this reliable upper-side data. Notice that the first term becomes zero when all of the data points are below $f$ (i.e., $y' \leq f_t(\boldsymbol{x}), \forall (x, y') \in \mathcal{D}'$), since then $\{\boldsymbol{X}_{\mathrm{up}}, \boldsymbol{y}'_{\mathrm{up}}\}$ becomes empty. The second term thus has the effect of pushing down $f$ at all of the data points so that some data points are above $f$. Meanwhile, the third term partially cancels this effect of the second term for the upper-side data to control the balance between the first and the second terms.

## 3.2 Unbiasedness and Consistency of Gradient

U2 regression is the learning algorithm based on the gradient, $\nabla\hat{\mathcal{L}}(f_t)$, in Equation 10 and uses only asymmetrically-corrupted data $\mathcal{D}'$. The use of $\nabla\hat{\mathcal{L}}(f_t)$ can be justified as follows:

**Proposition 3.2.** *Suppose that Assumption 2.1 holds and the loss function $L(f(\boldsymbol{x}), y)$ satisfies Condition 3.1. Then, the gradient $\nabla\tilde{\mathcal{L}}(f_t)$ in Equation 9 and its empirical approximation $\nabla\hat{\mathcal{L}}(f_t)$ in Equation 10 are unbiased and consistent with the gradient $\nabla\mathcal{L}(f_t)$ in Equation 6 a.s.*

*Proof.* We outline a proof here and provide a complete one in Appendix A.2. First, we rewrite Equation 7 into a gradient that only contains the expectation over $p(\boldsymbol{x}, y | f_t(\boldsymbol{x}) \leq y)$ with Condition 3.1. Then, we apply Lemma 2.2 to the gradient, and it becomes an expression identical to Equation 9. $\qquad\square$

In other words, U2 regression asymptotically produces the same result as the learning algorithm based on the gradient $\nabla\mathcal{L}(f_t)$ in Equation 6, which requires the uncorrupted data without incomplete observations, $\mathcal{D}$. The convergence rate of U2 regression is of the order $\mathcal{O}_p(1/\sqrt{n_{\mathrm{up}}} + 1/\sqrt{N})$ in accordance with the central limit theorem (Chung, 1968), where $\mathcal{O}_p$ denotes the order in probability.

We further justify our approach of having the specific form of Equation 9 by showing that a straightforward variant that uses $\mathcal{D}'$ as if it does not involve incomplete observations (i.e., $p(\boldsymbol{x}, y) \approx p(\boldsymbol{x}, y')$) can fail for our problem. To this end, we introduce an additional assumption on the observation process:

**Assumption 3.3.** *Assume $\epsilon_{\mathrm{a}} \perp\!\!\!\perp \boldsymbol{x}$.*

Then, we have

**Lemma 3.4.** *Let $\nabla\check{\mathcal{L}}(f_t)$ be a variant of the gradient in Equation 7 replacing $p(\boldsymbol{x}, y)$ with $p(\boldsymbol{x}, y')$, $\delta$ be the difference between the expectations of the gradients in the upper side and the lower side $\delta \equiv \left|\mathbb{E}_{p(\boldsymbol{x}, y | f(\boldsymbol{x}) \leq y)}[\nabla L(f(\boldsymbol{x}), y)] - \mathbb{E}_{p(\boldsymbol{x}, y | y < f(\boldsymbol{x}))}[\nabla L(f(\boldsymbol{x}), y)]\right|$, $0 < \eta < 1$ be probability when $0 \leq \epsilon_{\mathrm{s}}$, and $0 < \xi < 1$ be probability when $\epsilon_{\mathrm{a}} = 0$. Then, $\nabla\check{\mathcal{L}}(f_t)$ is not consistent with the gradient in Equation 6 a.s., and the difference (bias) between them at step $t + 1$ in the gradient descent is*

$$\frac{\eta(1 - \eta)(1 - \xi)}{(1 - \eta) + \eta(1 - \xi)}\delta \leq |\nabla\check{\mathcal{L}}(f_t) - \nabla\mathcal{L}(f_t)|. \tag{11}$$

*Proof.* We outline a proof here and provide a complete one in Appendix A.3. We first show that the bias $|\nabla\check{\mathcal{L}}(f_t) - \nabla\mathcal{L}(f_t)|$ can be represented by the difference between the expectation of $\boldsymbol{g}(f_t(\boldsymbol{x}))$

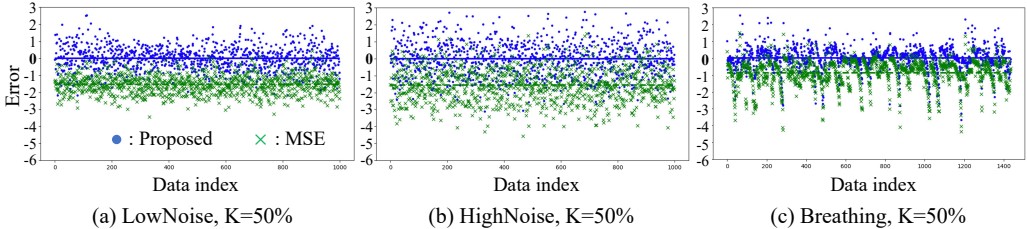

(a) LowNoise, K=50%          (b) HighNoise, K=50%          (c) Breathing, K=50%

Figure 2: Errors (predicted value minus true value) by proposed method (blue) and by `MSE` (green) for three tasks; (a) **LowNoise**, (b) **HighNoise**, and (c) **Breathing**, with $K = 50\%$ of incomplete training samples. Error of each data-point is shown by dot (for proposed method) or by cross mark (for `MSE`), and average error is shown by solid line (for proposed method) or by dashed line (for `MSE`).

with the upper-side data and that with the lower-side data, which can be written by $\delta$. The bias also has the coefficient which contains the proportions for the lower-side data and the original upper-side data mixed into the lower side due to incomplete observations. These values can be written by $\eta$ and $\xi$ from their definitions. □

Lemma 3.4 shows that the bias caused by incomplete observations becomes severe when there is a large difference between the expectations of the gradients in the upper side and the lower side. $\delta$ is usually higher than zero because $\delta = 0$ implies there is no difference between the expectations of the gradients in the upper side and the lower side or both of the expectations are zero. Furthermore, a larger $1 - \xi = p(\epsilon_a < 0)$ makes the bias more significant, which agrees with the intuition that as the proportion of incomplete observations increases, the problem becomes more difficult.

## 4   EXPERIMENTS

We now evaluate the proposed method through numerical experiments. We first introduce the baselines to be compared in the experiments. Then, we present the experimental results to show the effectiveness of our unbiased learning.

**Baselines.**   Recall that the novelty of the proposed approach lies in the unbiased gradient in Equation 10, which is derived from the new class of loss functions in equation 9 with Condition 3.1. An objective of our experiments is thus to validate the effectiveness of this new class of loss functions and the corresponding gradients against common loss functions in the literature. Specifically, we compare the proposed method with **MSE** (mean squared error), **MAE** (mean absolute error), and **Huber** losses (Huber et al., 1964; Narula & Wellington, 1982; Wilcox, 1997). For robust loss function in regression, MAE and Huber losses are considered the de facto standard and state-of-the-art in many studies and libraries. We use the same model and optimization method with all of the loss functions under consideration, and hence the only difference among the proposed method and the baselines is in the gradients. Since the loss function uniquely determines the baseline, we refer to each baseline method as `MSE`, `MAE`, or `Huber`.

### 4.1   EXPERIMENTAL PROCEDURE AND RESULTS

The experiments are organized into three parts. In Section 4.1.1, we visually demonstrate the effectiveness of the proposed approach in giving unbiased prediction. In Section 4.1.2, we intensively and quantitatively evaluate the predictive error of the proposed method and baselines with five real-world regression tasks. In Section 4.1.3, we demonstrate the practical benefit of our approach in a real healthcare use case, which has motivated this work. See the appendix for the details of the experimental settings.

#### 4.1.1   DEMONSTRATION OF UNBIASED LEARNING

**Procedure.**   We start by conducting the experiments with synthetic data to show the effectiveness of our method in obtaining unbiased learning results from asymmetrically-corrupted data with

Table 1: Comparison between proposed method and baselines in terms of MAE (smaller is better). Best methods are in bold. Confidence intervals are standard errors.

| | Specification | Throwing A | Lifting | Lowering | Throwing B | Avg. |
|---|---|---|---|---|---|---|
| MSE | 2.38±0.03 | 1.54±0.01 | 1.42±0.01 | 1.37±0.01 | 1.21±0.01 | 1.58 |
| MAE | 2.14±0.02 | 1.46±0.01 | 1.44±0.01 | 1.33±0.01 | 1.31±0.01 | 1.54 |
| Huber | 2.04±0.02 | 1.66±0.01 | 1.45±0.01 | 1.50±0.01 | 1.32±0.01 | 1.59 |
| Proposed-1 | 1.55±0.02 | 1.18±0.01 | 1.11±0.01 | 1.14±0.01 | 1.03±0.01 | 1.20 |
| Proposed-2 | **1.32±0.01** | **0.99±0.01** | **0.94±0.01** | **0.86±0.01** | **0.97±0.01** | **1.02** |

Table 2: Proportion of correct prediction period and rate of false prediction in real use case for healthcare. We estimate intrusive sensor output from outputs of non-intrusive sensors.

| | |
|---|---|
| Proportion of correct prediction period | 0.89 |
| Rate of false prediction | 0.016 |

different proportions of incomplete observations, $K = \{25, 50, 75\}\%$. We use three synthetic tasks, **LowNoise**, **HighNoise**, and **Breathing** collected from the Kaggle dataset (Sen, 2016). We compare the proposed method against MSE, which assumes that both upper- and lower-side data are correctly labeled. This comparison shows whether our method can learn from asymmetrically-corrupted data in an unbiased manner, which MSE cannot do.

**Results.** In Fig. 2, we plot the error in prediction (i.e., the predicted value minus the true value) given by the proposed method and MSE for each data-point of the three tasks with $K = 50\%$. Note that, for evaluating the unbiasedness, these test sets do not have incomplete observations. Since MSE regards both upper- and lower-side data as correctly labeled, it produces biased results due to the incomplete observations, where the average error (shown by the green dashed line) is negative, which means the estimation has a negative bias. In contrast, the average error by the proposed method (shown by the blue solid line) is approximately zero. This clearly shows that the proposed method obtained unbiased learning results. The figures for the other settings and tables showing quantitative performance are in Appendix E.

### 4.1.2 PERFORMANCE COMPARISON AMONG DIFFERENT LOSS FUNCTIONS

**Procedure.** We next apply the proposed method and baselines to five different real-world healthcare tasks from the UCI Machine Learning Repository (Velloso, 2013; Velloso et al., 2013) to show a more extensive comparison between the proposed method and the baselines (MSE, MAE, and Huber). For the proposed method, we use two implementations of $L(f(\boldsymbol{x}), y)$ for $f(\boldsymbol{x}) \leq y'$ in Equation 10: the absolute loss (Proposed-1) and the squared loss (Proposed-2). Here, we report the *mean absolute error* (MAE), and its standard error, of the predictions $\hat{\boldsymbol{y}} = \{\hat{y}_n\}_{n=1}^N$ against the corresponding true labels $\boldsymbol{y}$ across 5-fold cross-validation, each with a different randomly sampled training-testing split. MAE is the common metric used in the healthcare domain (Lee et al., 2016; Yeung et al., 2002; Wang et al., 2005; Srinivas et al., 2020) and is defined as $\mathrm{MAE}(\boldsymbol{y}, \hat{\boldsymbol{y}}) \equiv 1/N \sum_{n=1}^N |y_n - \hat{y}_n|$. For each fold of the cross-validation, we use a randomly sampled 20% of the training set as a validation set to choose the best hyperparameters for each algorithm, in which hyperparameters providing the highest MAE in the validation set are chosen.

**Results.** As seen in Table 1, Proposed-1 and Proposed-2 largely outperformes the baselines. The robust regression methods (MAE and Huber) did not improve in performance against MSE. In particular, Proposed-1 and Proposed-2 respectively reduced the MAE by more than 20% and 30% on average, compared with baselines.

### 4.1.3 REAL USE CASE FOR HEALTHCARE

**Procedure** Finally, we demonstrate the practicality of our approach in a real use case in healthcare. From non-intrusive bed sensors installed under each of the four legs of a bed, we estimate the motion intensity of a subject that could be measured accurately but intrusively with ActiGraph, a gold

standard sensor wrapped around the wrist (Tryon, 2013; Mullaney et al., 1980; Webster et al., 1982; Cole et al., 1992). If we can mimic the outputs from ActiGraph with outputs from the bed sensors, we can measure the motion with high accuracy and high coverage, while also easing the burden on the subject. We divide the dataset into three pieces and evaluate the results with 3-fold cross-validation. We here use the evaluation metrics that are specifically designed for sleep-wake discrimination (Cole et al., 1992) i.e., proportion of correct prediction period and rate of false prediction.

**Results.** Table 2 shows the proportion of correct prediction period and rate of false prediction, which indicate that the proposed method captured 89 percent of the total time period of the motions that were captured by ActiGraph, and false detection due to factors such as floor vibration was only 1.6 percent. Furthermore, the proposed method captured 15 additional motions that were not captured by ActiGraph. The baseline method `MSE` was severely underfitted, and most of the weights were zero; thus, we omitted these results. Overall, our findings here demonstrate that ActiGraph can be replaced with bed sensors, and we can also use the bed sensors for the inputs of certain ActiGraph functions, such as sleep-wake discrimination (Cole et al., 1992). See also Appendix G for further details, including the actual estimation results of the motion intensity.

## 5 DISCUSSION

**Limitations.** In this paper, we do not address symmetric label corruption, such as ordinary outliers, where the coverage and incompleteness are consistent between a label and explanatory variables. Other established approaches can handle such cases. Only when the corruption is asymmetric does it lead to the technical challenge we address here. In that sense, we can handle the opposite asymmetric corruption, in which labels for some observations may become inconsistently *higher* than those for typical observations. This can be handled as learning from *lower-side labeled data and unlabeled data*, i.e., LU regression. Since our derivation of U2 regression is straightforwardly applicable to this LU regression case, we show only its learning algorithm in Appendix C.

**Asymmetric Label Corruption in Classification.** In the classification problem setting, asymmetric label corruption is addressed with positive-unlabeled (PU) learning, where it is assumed that negative data cannot be obtained, but unlabeled data are available as well as positive data (Denis, 1998; De Comité et al., 1999; Letouzey et al., 2000; Shi et al., 2018; Kato et al., 2019; Sakai & Shimizu, 2019; Li et al., 2019; Zhang et al., 2019; 2020; Chen et al., 2020b;a; Luo et al., 2021; Hu et al., 2021; Li et al., 2021). An unbiased risk estimator has also been proposed (Du Plessis et al., 2014; 2015). However, PU classification cannot be used for a regression problem, where labels are real values and we need to handle order and gradation between labels. This is because its derivation and algorithm are based on the nature that labels must be binary, i.e., only positive or negative. We overcome this limitation with a novel approach based on an unbiased gradient.

**Future work.** We showed that our approach to estimating hyperparameters based on the grid search with the validation set was effective even for the one contains the important ratio for upper-side labeled data, $p(f_t(\boldsymbol{x}) \leq y)$. It also provides the flexibility needed to handle data variation. Most studies on PU learning assume that a hyperparameter corresponding to $\pi_{\mathrm{up}}$ is given (Hammoudeh & Lowd, 2020; Sonntag et al., 2021; Lin et al., 2022), and some papers have addressed this hyperparameter estimation as their main contribution (Jain et al., 2016; Ramaswamy et al., 2016; Christoffel et al., 2016; Jain et al., 2020; Yao et al., 2021). Developing a method for the hyperparameter estimation to improve performance would be a worthwhile next step of our study. Also, in Assumption 2.1, we assumed $\epsilon_{\mathrm{s}} \perp\!\!\!\perp f^*(\boldsymbol{x})$ and $\epsilon_{\mathrm{a}} \perp\!\!\!\perp f^*(\boldsymbol{x})$, which is a common noise assumption. Addressing the case when the noises are not independent of $f^*(\boldsymbol{x})$ is another future direction of our work.

**Conclusion.** We formulated a regression problem from asymmetrically-corrupted data in which training data are corrupted with an asymmetric noise that always has a negative value. This causes labels for data with relatively lower label values to be particularly unreliable. To address this problem, we proposed a learning algorithm, U2 regression. Under some technical assumptions, we showed that our algorithm is unbiased and consistent with regression that uses uncorrupted data without incomplete observations. Our analysis is based on the equivalence of the gradient between them. An experimental evaluation demonstrated that the proposed method was significantly better than the methods without the assumption of the asymmetrical label corruption.

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

# A  PROOFS

## A.1  PROOF OF LEMMA 2.2

*Proof.* For the proof of Lemma 2.2, we will derive two important lemmas from Assumption 2.1. Then, we will prove Lemma 2.2 by using them.

We first show $f^*(\boldsymbol{x}) \leq y' \Rightarrow \epsilon_a = 0$. When $f^*(\boldsymbol{x}) \leq y'$, we have from Equation 1 and Equation 4:

$$f^*(\boldsymbol{x}) \leq f^*(\boldsymbol{x}) + \epsilon_s + \epsilon_a \tag{12}$$
$$0 \leq \epsilon_s + \epsilon_a$$
$$-\epsilon_a \leq \epsilon_s.$$

Since $\epsilon_a \leq 0$ by Assumption 2.1, we have

$$|\epsilon_a| \leq \epsilon_s. \tag{13}$$

If $\epsilon_a < 0$, Assumption 2.1 implies $|\epsilon_s| < |\epsilon_a|$, which contradicts Equation 13. Hence, we must have

$$\epsilon_a = 0. \tag{14}$$

Since $y = y'$ when $\epsilon_a = 0$, we have

$$p(\boldsymbol{x}, y'|f^*(\boldsymbol{x}) \leq y') = p(\boldsymbol{x}, y'|f^*(\boldsymbol{x}) \leq y', \epsilon_a = 0) \tag{15}$$
$$= p(\boldsymbol{x}, y|f^*(\boldsymbol{x}) \leq y, \epsilon_a = 0)$$
$$= p(\boldsymbol{x}, y|f^*(\boldsymbol{x}) \leq y),$$

which establishes

**Lemma A.1.** *Let $p(\boldsymbol{x}, y, y')$ be the underlying probability distribution for $\boldsymbol{x}$, $y$, and $y'$. Then,*

$$p(\boldsymbol{x}, y'|f^*(\boldsymbol{x}) \leq y') = p(\boldsymbol{x}, y|f^*(\boldsymbol{x}) \leq y). \tag{16}$$

The condition parts of these conditional distributions represent the relationships between labels and regression functions, e.g., $p(\boldsymbol{x}, y|f^*(\boldsymbol{x}) \leq y)$ is the distribution of $x$ and $y$ when $y$ is higher than what is given by the oracle regression function $f^*$.

Similar to Lemma A.1, we show $f(x) \leq y' \Rightarrow \epsilon_a = 0$. Let $\mathcal{F}' \equiv \{f \in \mathcal{F} : |f(\boldsymbol{x}) - f^*(\boldsymbol{x})| \leq |\epsilon_s|$ a.s.$\}$, which represents our natural expectation that the regression function $f$ well approximates $f^*$. When $f(\boldsymbol{x}) \leq y'$, we have from Equation 1 and Equation 4 with the condition $f \in \mathcal{F}'$:

$$f(\boldsymbol{x}) \leq f^*(\boldsymbol{x}) + \epsilon_s + \epsilon_a \tag{17}$$
$$f(\boldsymbol{x}) \leq f(\boldsymbol{x}) + \epsilon_s + \epsilon_a + |\epsilon_s|$$
$$0 \leq \epsilon_s + \epsilon_a + |\epsilon_s|$$
$$-\epsilon_a \leq \epsilon_s + |\epsilon_s|.$$

Since $\epsilon_a \leq 0$ by Assumption 2.1, we have

$$|\epsilon_a| \leq \epsilon_s + |\epsilon_s|. \tag{18}$$

If $\epsilon_a < 0$, Assumption 2.1 implies $2|\epsilon_s| < |\epsilon_a|$, which contradicts Equation 18. Hence, we must have

$$\epsilon_a = 0. \tag{19}$$

Since $y = y'$ when $\epsilon_a = 0$, by replacing $f^*$ with $f$ for the argument in the derivation of Lemma A.1 in Equation 15, we have

**Lemma A.2.** *Let $\mathcal{F}' \equiv \{f \in \mathcal{F} : |f(\boldsymbol{x}) - f^*(\boldsymbol{x})| \leq |\epsilon_{\mathrm{s}}|\}$. When $f \in \mathcal{F}'$, the following holds:*

$$p(x, y'|f(\boldsymbol{x}) \leq y') = p(x, y|f(\boldsymbol{x}) \leq y). \tag{20}$$

Lemma A.1 immediately implies

$$\mathbb{E}_{p(\boldsymbol{x}, y'|f^*(\boldsymbol{x}) \leq y')}[G(\boldsymbol{x}, y')] = \mathbb{E}_{p(\boldsymbol{x}, y|f^*(\boldsymbol{x}) \leq y)}[G(\boldsymbol{x}, y)] \tag{21}$$

for any function $G : \mathbb{R}^D \times \mathbb{R} \to \mathbb{R}$ as long as the expectations exist. When $f \in \mathcal{F}'$, from Lemma A.2, we then have

$$\mathbb{E}_{p(\boldsymbol{x}, y'|f(\boldsymbol{x}) \leq y')}[G(\boldsymbol{x}, y')] = \mathbb{E}_{p(\boldsymbol{x}, y|f(\boldsymbol{x}) \leq y)}[G(\boldsymbol{x}, y)]. \tag{22}$$

$\square$

### A.2 PROOF OF PROPOSITION 3.2

*Proof.* From the decomposed gradients $\nabla\mathcal{L}(f_t)$ in Equation 7, we derive the proposed gradient only with the expectations over $p(\boldsymbol{x}, y')$.

From Condition 3.1 for $L(f(\boldsymbol{x}), y)$, $\nabla L(f(\boldsymbol{x}), y) = \boldsymbol{g}(f(\boldsymbol{x}))$ when $y < f(\boldsymbol{x})$. Thus, Equation 7 can be rewritten as

$$\nabla\mathcal{L}(f_t) = p(f_t(\boldsymbol{x}) \leq y)\mathbb{E}_{p(\boldsymbol{x}, y|f_t(\boldsymbol{x}) \leq y)}[\nabla L(f_t(\boldsymbol{x}), y)] \tag{23}$$
$$+ p(y < f_t(\boldsymbol{x}))\mathbb{E}_{p(\boldsymbol{x}|y < f_t(\boldsymbol{x}))}\Big[\boldsymbol{g}(f_t(\boldsymbol{x}))\Big],$$

where $y$ is marginalized out in the expectation in the second term since $\boldsymbol{g}(f_t(\boldsymbol{x}))$ does not depend on $y$.

Here, Equation 6 and Equation 7 can be rewritten by replacing $\nabla L(f_t(\boldsymbol{x}), y)$ with $\boldsymbol{g}(f_t(\boldsymbol{x}))$, as

$$\mathbb{E}_{p(\boldsymbol{x}, y)}[\boldsymbol{g}(f_t(\boldsymbol{x}))] = p(f_t(\boldsymbol{x}) \leq y)\mathbb{E}_{p(\boldsymbol{x}, y|f_t(\boldsymbol{x}) \leq y)}\Big[\boldsymbol{g}(f_t(\boldsymbol{x}))\Big] \tag{24}$$
$$+ p(y < f_t(\boldsymbol{x}))\mathbb{E}_{p(\boldsymbol{x}, y|y < f_t(\boldsymbol{x}))}\Big[\boldsymbol{g}(f_t(\boldsymbol{x}))\Big]$$
$$p(y < f_t(\boldsymbol{x}))\mathbb{E}_{p(\boldsymbol{x}, y|y < f_t(\boldsymbol{x}))}\Big[\boldsymbol{g}(f_t(\boldsymbol{x}))\Big] \tag{25}$$
$$= \mathbb{E}_{p(\boldsymbol{x}, y)}\Big[\boldsymbol{g}(f_t(\boldsymbol{x}))\Big] - p(f_t(\boldsymbol{x}) \leq y)\mathbb{E}_{p(\boldsymbol{x}, y|f_t(\boldsymbol{x}) \leq y)}\Big[\boldsymbol{g}(f_t(\boldsymbol{x}))\Big].$$

Since $\boldsymbol{g}(f_t(\boldsymbol{x}))$ does not depend on $y$, we can marginalize out $y$ in Equation 25 as

$$p(y < f_t(\boldsymbol{x}))\mathbb{E}_{p(\boldsymbol{x}|y < f_t(\boldsymbol{x}))}\Big[\boldsymbol{g}(f_t(\boldsymbol{x}))\Big] \tag{26}$$
$$= \mathbb{E}_{p(\boldsymbol{x})}\Big[\boldsymbol{g}(f_t(\boldsymbol{x}))\Big] - p(f_t(\boldsymbol{x}) \leq y)\mathbb{E}_{p(\boldsymbol{x}|f_t(\boldsymbol{x}) \leq y)}\Big[\boldsymbol{g}(f_t(\boldsymbol{x}))\Big].$$

From Equation 26, we can express Equation 23 as

$$\nabla\mathcal{L}(f_t) = p(f_t(\boldsymbol{x}) \leq y)\mathbb{E}_{p(\boldsymbol{x}, y|f_t(\boldsymbol{x}) \leq y)}[\nabla L(f_t(\boldsymbol{x}), y)] \tag{27}$$
$$+ \mathbb{E}_{p(\boldsymbol{x})}\Big[\boldsymbol{g}(f_t(\boldsymbol{x}))\Big] - p(f_t(\boldsymbol{x}) \leq y)\mathbb{E}_{p(\boldsymbol{x}|f_t(\boldsymbol{x}) \leq y)}\Big[\boldsymbol{g}(f_t(\boldsymbol{x}))\Big].$$

Finally, from Lemma 2.2, we can rewrite Equation 27 as:

$$\nabla\mathcal{L}(f_t) = p(f_t(\boldsymbol{x}) \leq y)\mathbb{E}_{p(\boldsymbol{x}, y'|f_t(\boldsymbol{x}) \leq y')}[\nabla L(f_t(\boldsymbol{x}), y)] \tag{28}$$
$$+ \mathbb{E}_{p(\boldsymbol{x})}\Big[\boldsymbol{g}(f_t(\boldsymbol{x}))\Big] - p(f_t(\boldsymbol{x}) \leq y)\mathbb{E}_{p(\boldsymbol{x}|f_t(\boldsymbol{x}) \leq y')}\Big[\boldsymbol{g}(f_t(\boldsymbol{x}))\Big],$$

which is identical to Equation 9. Thus, the gradient in Equation 9 is unbiased and consistent with the gradient in Equation 6 a.s. $\square$

## A.3 PROOF OF LEMMA 3.4

*Proof.* The difference between the decomposed gradients $\nabla \check{\mathcal{L}}(f_t)$ and $\nabla \mathcal{L}(f_t)$ at step $t+1$ in the gradient descent is

$$|\nabla \check{\mathcal{L}}(f_t) - \nabla \mathcal{L}(f_t)| \tag{29}$$

$$= \Bigg| p(f_t(\boldsymbol{x}) \le y) \mathbb{E}_{p(\boldsymbol{x}, y' | f_t(\boldsymbol{x}) \le y')} [\nabla L(f_t(\boldsymbol{x}), y)]$$

$$+ p(y < f_t(\boldsymbol{x})) \mathbb{E}_{p(\boldsymbol{x}, y' | y' < f_t(\boldsymbol{x}))} [\nabla L(f_t(\boldsymbol{x}), y)]$$

$$- p(f_t(\boldsymbol{x}) \le y) \mathbb{E}_{p(\boldsymbol{x}, y | f_t(\boldsymbol{x}) \le y)} [\nabla L(f_t(\boldsymbol{x}), y)]$$

$$- p(y < f_t(\boldsymbol{x})) \mathbb{E}_{p(\boldsymbol{x}, y | y < f_t(\boldsymbol{x}))} [\nabla L(f_t(\boldsymbol{x}), y)] \Bigg|.$$

From Lemma 2.2 and Condition 3.1,

$$|\nabla \check{\mathcal{L}}(f_t) - \nabla \mathcal{L}(f_t)| \tag{30}$$

$$= \Bigg| p(y < f_t(\boldsymbol{x})) \mathbb{E}_{p(\boldsymbol{x}, y' | y' < f_t(\boldsymbol{x}))} [\nabla L(f_t(\boldsymbol{x}), y)]$$

$$- p(y < f_t(\boldsymbol{x})) \mathbb{E}_{p(\boldsymbol{x}, y | y < f_t(\boldsymbol{x}))} [\nabla L(f_t(\boldsymbol{x}), y)] \Bigg|$$

$$= \Bigg| p(y < f_t(\boldsymbol{x})) \mathbb{E}_{p(\boldsymbol{x} | y' < f_t(\boldsymbol{x}))} [\boldsymbol{g}(f_t(\boldsymbol{x}))]$$

$$- p(y < f_t(\boldsymbol{x})) \mathbb{E}_{p(\boldsymbol{x} | y < f_t(\boldsymbol{x}))} [\boldsymbol{g}(f_t(\boldsymbol{x}))] \Bigg|.$$

We decompose $\mathbb{E}_{p(\boldsymbol{x} | y' < f_t(\boldsymbol{x}))} [\boldsymbol{g}(f_t(\boldsymbol{x}))]$ again as

$$|\nabla \check{\mathcal{L}}(f_t) - \nabla \mathcal{L}(f_t)| \tag{31}$$

$$= \Bigg| p(y < f_t(\boldsymbol{x})) \Bigg($$

$$p(f_t(\boldsymbol{x}) \le y | y' < f_t(\boldsymbol{x})) \mathbb{E}_{p(\boldsymbol{x} | y' < f_t(\boldsymbol{x}) \wedge f_t(\boldsymbol{x}) \le y)} [\boldsymbol{g}(f_t(\boldsymbol{x}))]$$

$$+ p(y < f_t(\boldsymbol{x}) | y' < f_t(\boldsymbol{x})) \mathbb{E}_{p(\boldsymbol{x} | y' < f_t(\boldsymbol{x}) \wedge y < f_t(\boldsymbol{x}))} [\boldsymbol{g}(f_t(\boldsymbol{x}))] \Bigg)$$

$$- p(y < f_t(\boldsymbol{x})) \mathbb{E}_{p(\boldsymbol{x} | y < f_t(\boldsymbol{x}))} [\boldsymbol{g}(f_t(\boldsymbol{x}))] \Bigg|.$$

The condition $y' < f_t(x) \wedge y < f_t(x)$ is equivalent to the condition $y < f_t(x)$ since $y' \le y$ from Assumption 2.1 and thus $p(y' < f_t(x) | y < f_t(x)) = 1$. Then, we have

$$|\nabla \check{\mathcal{L}}(f_t) - \nabla \mathcal{L}(f_t)| \tag{32}$$

$$= \Bigg| p(y < f_t(\boldsymbol{x})) \Bigg($$

$$p(f_t(\boldsymbol{x}) \le y | y' < f_t(\boldsymbol{x})) \mathbb{E}_{p(\boldsymbol{x} | y' < f_t(\boldsymbol{x}) \wedge f_t(\boldsymbol{x}) \le y)} [\boldsymbol{g}(f_t(\boldsymbol{x}))]$$

$$+ p(y < f_t(\boldsymbol{x}) | y' < f_t(\boldsymbol{x})) \mathbb{E}_{p(\boldsymbol{x} | y < f_t(\boldsymbol{x}))} [\boldsymbol{g}(f_t(\boldsymbol{x}))] \Bigg)$$

$$- p(y < f_t(\boldsymbol{x})) \mathbb{E}_{p(\boldsymbol{x} | y < f_t(\boldsymbol{x}))} [\boldsymbol{g}(f_t(\boldsymbol{x}))] \Bigg|.$$

Additionally, since $p(y < f_t(\boldsymbol{x}) | y' < f_t(\boldsymbol{x})) = 1 - p(f_t(\boldsymbol{x}) \le y | y' < f_t(\boldsymbol{x}))$,

$$|\nabla \check{\mathcal{L}}(f_t) - \nabla \mathcal{L}(f_t)| \tag{33}$$

$$= \Bigg| p(y < f_t(\boldsymbol{x})) p(f_t(\boldsymbol{x}) \le y | y' < f_t(\boldsymbol{x})) \Bigg($$

$$\mathbb{E}_{p(\boldsymbol{x} | y' < f_t(\boldsymbol{x}) \wedge f_t(\boldsymbol{x}) \le y)} [\boldsymbol{g}(f_t(\boldsymbol{x}))] - \mathbb{E}_{p(\boldsymbol{x} | y < f_t(\boldsymbol{x}))} [\boldsymbol{g}(f_t(\boldsymbol{x}))] \Bigg) \Bigg|.$$

This equation shows that the bias is represented by the difference between the expectation of $g(f_t(x))$ with the lower-side data and that with the original upper-side data mixed into the lower side due to incomplete observations and the corresponding proportions.

From Assumption 3.3, since $\epsilon_a \perp\!\!\!\perp x$,

$$|\nabla \check{\mathcal{L}}(f_t) - \nabla \mathcal{L}(f_t)| \tag{34}$$

$$= \left| p(y < f_t(x)) p(f_t(x) \le y | y' < f_t(x)) \Bigg( \right.$$

$$\left. \mathbb{E}_{p(x|f_t(x) \le y)}[g(f_t(x))] - \mathbb{E}_{p(x|y < f_t(x))}[g(f_t(x))] \Bigg) \right|.$$

Since $|f - f^*| \le |\epsilon_s|$ a.s., $p(f_t(x) \le y) = \eta$ and $p(y < f_t(x)) = 1 - \eta$ from their definition,

$$p(f_t(x) \le y | y' < f_t(x)) = \frac{p(f_t(x) \le y) p(\epsilon_a < 0)}{p(y < f_t(x)) + p(f_t(x) \le y) p(\epsilon_a < 0)}$$

$$= \frac{\eta(1 - \xi)}{(1 - \eta) + \eta(1 - \xi)}. \tag{35}$$

Therefore, from the definition of $\delta$,

$$|\nabla \check{\mathcal{L}}(f_t) - \nabla \mathcal{L}(f_t)| \ge \frac{\eta(1 - \eta)(1 - \xi)}{(1 - \eta) + \eta(1 - \xi)} \delta. \tag{36}$$

$\square$

## B    IMPLEMENTATION OF LEARNING ALGORITHM BASED ON STOCHASTIC OPTIMIZATION

We scale up our U2 regression algorithm by stochastic approximation with $M$ mini-batches and add a regularization term, $R(f)$:

$$\nabla \hat{\mathcal{L}}^{\{m\}}(f_t) = \sum_{(x,y) \in \left\{ X_{\mathrm{up}}^{\{m\}}, y_{\mathrm{up}}^{\{m\}} \right\}} \nabla L(f_t(x), y) \tag{37}$$

$$+ \rho \Bigg[ \sum_{x \in X_{\mathrm{un}}^{\{m\}}} g(f_t(x)) \Bigg] - \sum_{x \in X_{\mathrm{up}}^{\{m\}}} g(f_t(x)) + \lambda \frac{\partial R(f_t)}{\partial \theta},$$

where $\nabla \hat{\mathcal{L}}^{\{m\}}(f_t)$ is the gradient for the $m$-th mini-batch, $\{X_{\mathrm{up}}^{\{m\}}, y_{\mathrm{up}}^{\{m\}}\}$ and $X_{\mathrm{un}}^{\{m\}}$ respectively are upper-side and unlabeled sets in the $m$-th mini-batch based on the current $f_t$, $\lambda$ is a regularization parameter, and the regularization term $R(f)$ is, for example, the L1 or L2 norm of the parameter vector $\theta$ of $f$. We also convert $n_{\mathrm{up}}/(\pi_{\mathrm{up}} N)$ to a hyperparameter $\rho$, ignoring constant coefficients instead of directly handling $\pi_{\mathrm{up}}$. The hyperparameters $\rho$ and $\lambda$ are optimized in training based on the grid-search with the validation set.

The U2 regression algorithm based on stochastic optimization is described in Algorithm 1. We learn the regression function with the gradient in Equation 37 by using any stochastic gradient method. Here, we used Adam with the hyperparameters recommended in Kingma & Ba (2015), and the number of samples in the mini-batches was set to 32. We set the candidates of the hyperparameters, $\rho$ and $\lambda$, to $\{10^{-3}, 10^{-2}, 10^{-1}, 10^{0}\}$. By using the learned $f$, we can estimate $\hat{y} = f(x)$ for new data $x$.

## C    ALGORITHM FOR LU REGRESSION

We show the algorithm for the *lower and unlabeled* regression (LU regression), where labels for some observations may become inconsistently *higher* than those for typical observations. Let $L_{\mathrm{LU}}(f(x), y)$ be a loss function for LU regression and $g_{\mathrm{LU}}(f(x))$ be a gradient $\nabla L_{\mathrm{LU}}(f(x), y)$ when $f(x) \le y$. Similar to Condition 3.1 for U2 regression, we assume that the class of $L_{\mathrm{LU}}(f(x), y)$ satisfies the

---

**Algorithm 1** U2 regression based on stochastic gradient method.

---

**Input:** Training data $\mathcal{D}' = \{\boldsymbol{x}_n, y'_n\}_{n=1}^N$; hyperparameters $\rho, \lambda \geq 0$; an external stochastic gradient method $\mathcal{A}$
**Output:** Model parameters $\boldsymbol{\theta}$ for $f$
 1: **while** No stopping criterion has been met
 2:      Shuffle $\mathcal{D}'$ into $M$ mini-batches: $\left\{\boldsymbol{X}^{\{m\}}, \boldsymbol{y}^{\{m\}}\right\}_{m=1}^M$
 3:      **for** $m = 1$ **to** $M$
 4:          Compute the gradient $\nabla\hat{\mathcal{L}}^{\{m\}}(f_t)$ in Equation 37 with $\left\{\boldsymbol{X}^{\{m\}}, \boldsymbol{y}^{\{m\}}\right\}$
 5:          Update $\boldsymbol{\theta}$ by $\mathcal{A}$ with $\nabla\hat{\mathcal{L}}^{\{m\}}(f_t)$

---

condition that $\boldsymbol{g}_{\mathrm{LU}}(f(\boldsymbol{x}))$ is a gradient function depending only on $f(\boldsymbol{x})$ and not on the value of $y$. Then, LU regression is Algorithm 1, with the following gradient, $\nabla\hat{\mathcal{L}}_{\mathrm{LU}}^{\{m\}}(f_t)$, instead of $\nabla\hat{\mathcal{L}}^{\{m\}}(f_t)$ in Equation 37, as

$$\nabla\hat{\mathcal{L}}_{\mathrm{LU}}^{\{m\}}(f_t) = \sum_{\{\boldsymbol{x},y\}\in\left\{\boldsymbol{X}_{\mathrm{lo}}^{\{m\}}, \boldsymbol{y}_{\mathrm{lo}}^{\{m\}}\right\}} \nabla L_{\mathrm{LU}}(f_t(\boldsymbol{x}), y) \tag{38}$$
$$+ \rho\left[\sum_{\boldsymbol{x}\in\boldsymbol{X}_{\mathrm{un}}^{\{m\}}} \boldsymbol{g}_{\mathrm{LU}}(f_t(\boldsymbol{x}))\right] - \sum_{\boldsymbol{x}\in\boldsymbol{X}_{\mathrm{lo}}^{\{m\}}} \boldsymbol{g}_{\mathrm{LU}}(f_t(\boldsymbol{x})) + \lambda\frac{\partial R(f_t)}{\partial\boldsymbol{\theta}},$$

where $\{\boldsymbol{X}_{\mathrm{lo}}^{\{m\}}, \boldsymbol{y}_{\mathrm{lo}}^{\{m\}}\}$ and $\boldsymbol{X}_{\mathrm{un}}^{\{m\}}$ respectively are lower-side and unlabeled sets in the $m$-th mini-batch based on the current $f_t$.

## D    COMPUTING INFRASTRUCTURE

All of the experiments were carried out with a Python and TensorFlow implementation on workstations having 80 GB of memory, a 4.0 GHz CPU, and an Nvidia Titan X GPU. In this environment, the computational time to produce the results was a few hours.

## E    DETAILS OF EXPERIMENTS IN SECTION 4.1.1

### E.1    SYNTHETIC DATASETS

We conducted the experiments on synthetic data to evaluate the feasibility of our method for obtaining unbiased learning results from asymmetrically-corrupted data containing different proportions of incomplete observations. We generated synthetic data on the basis of Assumption 2.1 and Equation 4. We randomly generated $N = 1000$ training samples, $\boldsymbol{X} = \{\boldsymbol{x}_n\}_{n=1}^N$, from the standard Gaussian distribution $\mathcal{N}(\boldsymbol{x}_n; 0, \boldsymbol{I})$, where the number of features in $\boldsymbol{x}$ was $D = 10$, and $\boldsymbol{I}$ is the identity matrix. Then, using $\boldsymbol{X}$, we generated the corresponding $N$ sets of true labels $\boldsymbol{y} = \{y_n\}_{n=1}^N$ from the distribution $\mathcal{N}(y_n; \boldsymbol{w}^\top\boldsymbol{x}_n, \beta)$, where $\boldsymbol{w}$ are coefficients that were also randomly generated from the standard Gaussian distribution $\mathcal{N}(\boldsymbol{w}; 0, \boldsymbol{I})$, $\beta$ is the noise precision, and $\top$ denotes the transpose. For simulating the situation in which a label has incomplete observations, we created corrupted labels $\boldsymbol{y}' = \{y'_n\}_{n=1}^N$ by randomly selecting $K$ percent of data in $\boldsymbol{y}$ and subtracting the absolute value of white Gaussian noise with twice the value of the precision as that of $\boldsymbol{y}$, $2\beta$, from their values. We repeatedly evaluated the proposed method for each of the following settings. The noise precision was $\beta = \{10^0, 10^{-1}\}$, which corresponded to a low-noise setting task (**LowNoise**) and a high-noise setting task (**HighNoise**), and the proportion of incomplete training samples was $K = \{25, 50, 75\}\%$. In the case of $K = 75\%$, only 25 percent of the samples correctly corresponded to labels, and all of the other samples were attached with labels that were lower than the corresponding true values. It is quite difficult to learn regression functions using such data. In these tasks, we used a linear model, $\boldsymbol{\theta}^\top\boldsymbol{x}$, for $f(\boldsymbol{x})$ and an implementation for Equation 37 with the absolute loss, which satisfies Condition 3.1, for the loss function $L$ and L1-regularization for the regularization term. We set the candidates of the hyperparameters, $\rho$ and $\lambda$, to $\{10^{-3}, 10^{-2}, 10^{-1}, 10^0\}$. We standardized the data by subtracting their mean and dividing by their standard deviation in the training split. We used Adam

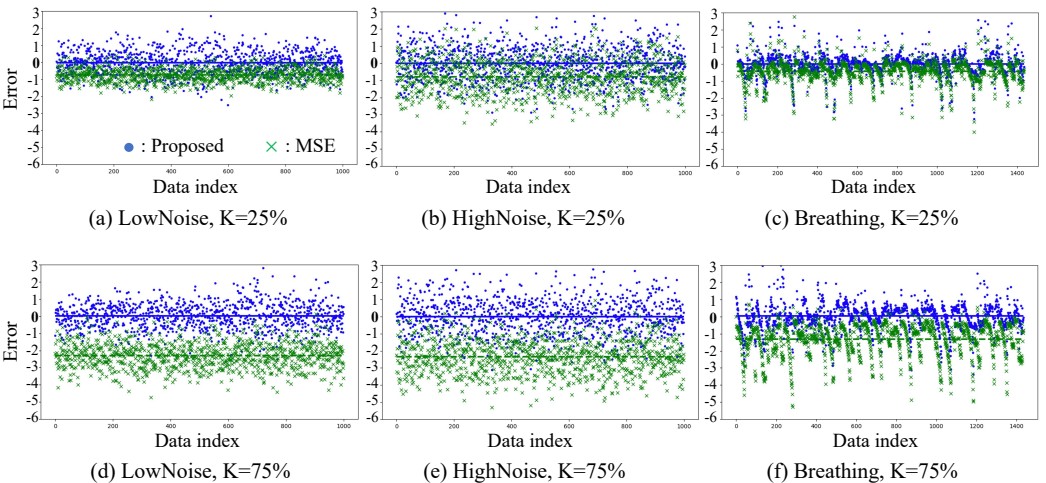

Figure 3: Errors in prediction (predicted value minus true value) by proposed method (blue) and by `MSE` (green) for tasks; (a) **LowNoise**, $K = 25\%$, (b) **HighNoise**, $K = 25\%$, (c) **Breathing**, $K = 75\%$, (d) **LowNoise**, $K = 75\%$, (e) **HighNoise**, $K = 75\%$, and (f) **Breathing**, $K = 75\%$. Error of each data-point is shown by dot (for proposed method) or by cross mark (for `MSE`), and average error is shown by solid line (for proposed method) or by dashed line (for `MSE`).

with the hyperparameters recommended in Kingma & Ba (2015), and the number of samples in the mini-batches was set to 32.

We also used a real-world sensor dataset collected from the Kaggle dataset (Sen, 2016) that contains breathing signals (**Breathing**). The dataset consisted of $N = 1,432$ samples. We used signals from a chest belt as $\boldsymbol{X} = \{\boldsymbol{x}_n\}_{n=1}^N$, and $\boldsymbol{x}$ in each sample had $D = 2$ number of features, i.e., the period and height of the expansion/contraction of the chest. We used signals obtained by the Douglas bag (DB) method, which is the gold standard for measuring ventilation, as true labels $\boldsymbol{y} = \{y_n\}_{n=1}^N$. For our problem setting, we created corrupted labels $\boldsymbol{y}' = \{y'_n\}_{n=1}^N$ through the same procedure for synthetic corruption as that for LowNoise and HighNoise with $K = \{25, 50, 75\}\%$. In the experiment on Breathing, for its non-linearity, we used $\boldsymbol{\theta}^\top \boldsymbol{\phi}(\boldsymbol{x}, \sigma)$ for $f(\boldsymbol{x})$, where $\phi$ is a radial basis function with the training set as its bases, and $\sigma$ is a hyperparameter representing the kernel width that is also optimized by using the validation set. We set the candidates of the hyperparameter $\sigma$ to $\{10^{-3}, 10^{-2}, 10^{-1}, 10^0\}$. The other implementation details were the same as those for **LowNoise** and **HighNoise**.

### E.2 DETAILED RESULTS

Figure 3 shows the error between the estimation results of the proposed method and their true values and those of `MSE` for **LowNoise**, **HighNoise**, and **Breathing** with 25 and 75 percent of incomplete training samples. Table 3 shows the performance on **LowNoise**, **HighNoise**, and **Breathing** for the proposed method and `MSE`. As shown in Figure 3, the proposed method obtains unbiased learning results in all cases, while `MSE` produces biased results. From Table 3, we can see that the proposed method outperformes `MSE` overall. We found that the performance of our method is not significantly affected by the increase in the proportion of incomplete training samples $K$ even for $K = 75\%$, unlike that of `MSE`.

### E.3 PERFORMANCE OVER DIFFERENT SIZES OF VALIDATION SET

To demonstrate the robustness of our validation-set-based approach to estimating the hyperparameter $\pi_{\mathrm{up}}$, we show the performance of the proposed method over different sizes of the validation set in Fig. 4. This analysis is conducted on the tasks in Section 4.1.1; LowNoise, HighNoise, and Breathing, with $K = 50\%$. Figure 4 shows that the proposed method does not degrade its performance much,

Table 3: Comparison of proposed method and `MSE` in terms of MAE (smaller is better). Best methods are in bold. Confidence intervals are standard errors.

| | LowNoise | | | HighNoise | | |
|---|---|---|---|---|---|---|
| | $K = 25\%$ | $K = 50\%$ | $K = 75\%$ | $K = 25\%$ | $K = 50\%$ | $K = 75\%$ |
| `MSE` | $0.77 \pm 0.01$ | $1.53 \pm 0.02$ | $2.30 \pm 0.02$ | $1.03 \pm 0.02$ | $1.62 \pm 0.03$ | $2.36 \pm 0.03$ |
| Proposed | $\mathbf{0.55 \pm 0.01}$ | $\mathbf{0.54 \pm 0.01}$ | $\mathbf{0.58 \pm 0.01}$ | $\mathbf{0.79 \pm 0.02}$ | $\mathbf{0.80 \pm 0.02}$ | $\mathbf{0.80 \pm 0.02}$ |

| | Breathing | | |
|---|---|---|---|
| | $K = 25\%$ | $K = 50\%$ | $K = 75\%$ |
| `MSE` | $0.55 \pm 0.02$ | $0.91 \pm 0.02$ | $1.32 \pm 0.02$ |
| Proposed | $\mathbf{0.43 \pm 0.01}$ | $\mathbf{0.46 \pm 0.01}$ | $\mathbf{0.59 \pm 0.01}$ |

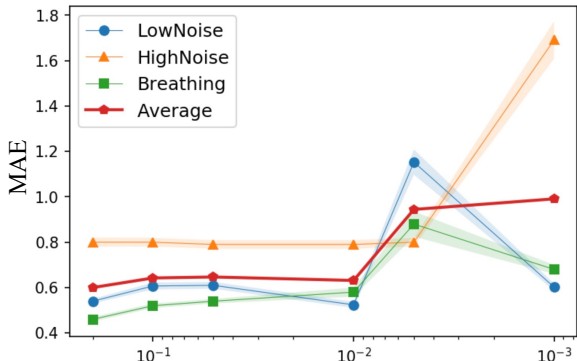

Figure 4: Performance (MAE, lower is better) of proposed method over different sizes of validation set. Blue line represents results on LowNoise task, orange line represents results on HighNoise task, green line represents results on Breathing task, with $K = 50\%$ of incomplete training samples, and red line is their average. Shaded areas are confidence intervals. Leftmost point is results when we use $20\%$ of training set as validation set, which is setting we used in experiments throughout this paper, and rightmost point is those of $0.1\%$.

even when we use only $1\%$ of the training set as the validation set. This demonstrates that the proposed approach is robust enough also for the small size of the validation set as well as the high proportion of incomplete validation samples. In Fig. 5, we also show a chart similar to Fig. 2 (the error in prediction) when we used $1\%$ of the training set as the validation set. We can see that even in this case, the proposed method achieved unbiased learning (the average error shown by the blue solid line is approximately zero.).

## F DETAILS OF EXPERIMENTS IN SECTION 4.1.2

We applied the algorithm to five different real-world healthcare tasks recorded in the datasets from the UCI Machine Learning Repository (Velloso, 2013; Velloso et al., 2013), which contains sensor outputs from wearable devices attached to the arm while subjects exercised. From the non-intrusive sensors attached to gym equipment, we estimated the motion intensity of a subject that was measured accurately with an arm sensor that was an intrusive sensor wrapped around the arm. If we can mimic outputs from the arm sensor with outputs from the equipment sensor, it could contribute to the subjects' comfort, as they would not need to wear sensors to measure their motion intensity. We used all of the features from the equipment sensor that took "None" values less than ten times as $\boldsymbol{X} = \{\boldsymbol{x}_n\}_{n=1}^N$, where each sample had $D = 13$ number of features. The corrupted labels $\boldsymbol{y}' = \{y'_n\}_{n=1}^N$ were the magnitude of acceleration from the arm sensor, which can accurately sense motion intensity on the arm, but it had insufficient data coverage and incomplete or missing observations for the movements of other body parts. For performance evaluation, we used the magnitude of acceleration for the

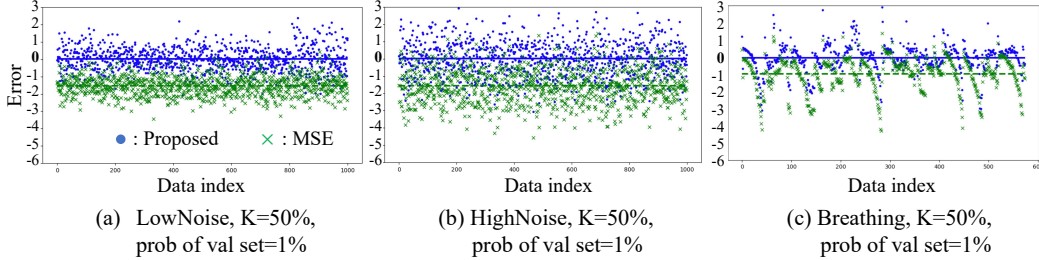

(a) LowNoise, K=50%,
prob of val set=1%

(b) HighNoise, K=50%,
prob of val set=1%

(c) Breathing, K=50%,
prob of val set=1%

Figure 5: Errors in prediction when we choose hyperparameters with $1\%$ of training set as validation set. Tasks are (a) **LowNoise**, (b) **HighNoise**, and (c) **Breathing**, with $K = 50\%$ of incomplete training samples. Error of each data-point is shown by blue dot (for proposed method) or by green cross mark (for MSE), and average error is shown by blue solid line (for proposed method) or by green dashed line (for MSE).

entire body as true labels $\boldsymbol{y} = \{y_n\}_{n=1}^N$. The number of samples were $N = 11,159$, $N = 7,593$, $N = 6,844$, $N = 6,432$, and $N = 7,214$ respectively for the tasks, Specification, Throwing A, Lifting, Lowering, and Throwing B. For the complex nature of the tasks, we used a 6-layer multilayer perceptron with ReLU (Nair & Hinton, 2010) (more specifically, $D$-100-100-100-100-1) as $f(\boldsymbol{x})$, which demonstrates the usefulness of the proposed method for training deep neural networks. We also used a dropout (Srivastava et al., 2014) with a rate of $50\%$ after each fully connected layer. We used two implementations for $L(f(\boldsymbol{x}), y)$ when $f(\boldsymbol{x}) \leq y'$ in Equation 37 with the absolute loss (`Proposed-1`) and the squared loss (`Proposed-2`). For both implementations, we used the absolute loss, which satisfies Condition 3.1, for the loss function $L(f(\boldsymbol{x}), y)$ when $y' < f(\boldsymbol{x})$ and used L1-regularization for the regularization term. The other implementation details were the same as those for **LowNoise**, **HighNoise**, and **Breathing**.

## G    DETAILS OF EXPERIMENTS IN SECTION 4.1.3

We demonstrate the practicality of our approach in a real use case in healthcare. From non-intrusive bed sensors installed under each of the four legs of a bed, we estimated the motion intensity of a subject that was measured accurately with ActiGraph, a gold standard intrusive sensor wrapped around the wrist (Tryon, 2013; Mullaney et al., 1980; Webster et al., 1982; Cole et al., 1992). The sensing results of ActiGraph are used for tasks such as discriminating whether a subject is asleep or awake (Cole et al., 1992). While ActiGraph can accurately sense motion on the forearm, it has insufficient data coverage in other areas and often causes observations of movements on other body parts to be missing. The bed sensors have a broader data coverage since they can sense global motion on all body parts; however, the sensing accuracy is limited due to their non-intrusiveness. If we can mimic the outputs from ActiGraph with outputs from the bed sensors, we can expect to achieve sufficient accuracy and coverage while also easing the burden on the subject. The dataset we used included three pieces of data, Data (i), (ii), and (iii), which were respectively recorded over 20, 18, and 18.5 minutes. Each piece of data consisted of pairs of bed-sensor-data sequences and the corresponding motion intensity sequence obtained by ActiGraph. We used the "magnitude" attribute of ActiGraph as corrupted labels $\boldsymbol{y}'$ for the motion intensity, whose sampling rate was about one sample per second. For true labels $\boldsymbol{y}$, we manually measured the motion intensity every minute under the management of a domain expert. For $\boldsymbol{X}$, we first computed the gravity center of the four sensor outputs that were obtained from the bed sensors under the four legs of a bed. Then, we computed the time derivatives and cross terms of the raw sensor outputs and the gravity center. The sampling rate of the bed sensors was different from that of ActiGraph, about one sample per five milliseconds. Thus, $\boldsymbol{X}$ was finally generated as a sliding window of statistics in $1,000$-millisecond (1 second) subsequences of the time series of the above computed variables, where 1 second was the same as the sampling interval of ActiGraph. The statistics were means, standard deviations, and $\{0.05, 0.25, 0.5, 0.75, 0.95\}$ quantiles. In this task, we used the linear model $\boldsymbol{\theta}^\top \boldsymbol{x}$ for $f(\boldsymbol{x})$ due to its interpretability, which is inevitable in real-world healthcare and medical applications.

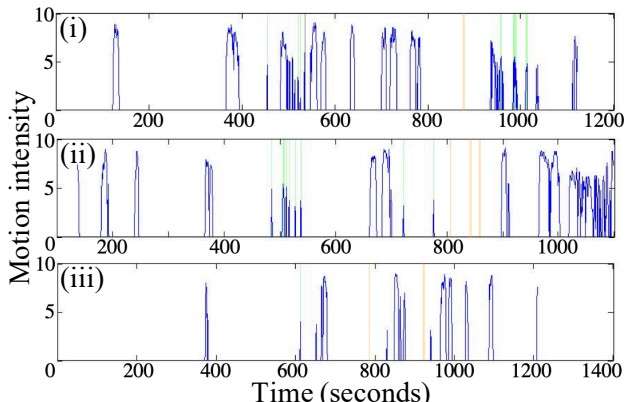

Figure 6: Experiment on real use case for healthcare. Blue line represents our estimation results for motion intensity. White (non-colored) area shows that both proposed method and ActiGraph correctly estimated motion intensity of the subject at this duration. Green area shows that our method could capture motion at this duration while ActiGraph could not. Orange area shows that our method could not capture motion at this duration but ActiGraph could. Gray area shows that our method mistakenly captured noise as subject's motion.

### G.1 ESTIMATION RESULTS FOR MOTION INTENSITY

Figure 6 compares our estimation results for motion intensity with the output of ActiGraph and true labels.

### G.2 IMPORTANT FEATURES ESTIMATING MOTION INTENSITY

The important features selected by L1 regularization were the statistics of the gravity center and the cross terms and time derivatives of the raw sensor outputs. The largest weight was assigned to the standard deviation of the gravity center, which represents the amplitude of the gravity center, so it is directly related to the motion of subjects.

## H OTHER POSSIBLE USE CASES OF REGRESSION FOR SENSOR MAGNITUDE

Examples of predicting the magnitude values of a sensor, which is a field of application of U2 regression, can be found in several areas. Besides the medical and healthcare applications discussed in the main text, another example is estimating the wind speed or rainfall in a specific region from observable macroscopic information (Cheng & Tan, 2008; Abraham & Tan, 2010; Abraham et al., 2013; Vandal et al., 2017), known as statistical downscaling (Wilby et al., 2004). Wind speed and rainfall, which are labels in these tasks, can be sensed locally in a limited number of locations and provide incomplete observations and biased labels compared with macroscopic information, which is considered to be explanatory variables.

