# OpenReview forum: "Learning from Asymmetrically-corrupted Data in Regression for Sensor Magnitude"
_ICLR.cc/2023/Conference — Submitted to ICLR 2023_

### Official Review · Reviewer_mxCJ · 2022-10-22

**Confidence:** 4
**Correctness:** 3
**Technical Novelty And Significance:** 3
**Empirical Novelty And Significance:** 4
**Recommendation:** 6

**Clarity, Quality, Novelty And Reproducibility:**

The introduction and problem statement are clear, the paper provides a novel solution with some theoretical analysis. The experimental setting is simple and well-explained, so I believe it is reproducible.


**Strength And Weaknesses:**

Strengths: The paper is well written, and the English level is satisfactory. The proposed problem is interesting with a novel solution. They prove the unbiasedness and consistency of the gradient of the proposed loss function. Several examples are provided to demonstrate the advantages of the proposed approach.

Weaknesses: The authors only compare their method to simple baselines, while several solutions exist to the related problem of regression with asymmetric loss functions.


Minor comments:
For clarity, please mention what K is in the caption of figure 2.

Page 4 results of figure 2, I assume that the data in the test set is not incomplete, but this is not explained.

Please expand on the caption of Table 2.

P9 the results related to table 2, can be viewed as a classification task, so why not compare to baselines that focus on asymmetric classification?

Why is the word Rate capitalized?



**Summary Of The Paper:**


The authors address an interesting problem of the asymmetrically corrupted regression problem. They motivate the problem using several real-world examples. They propose a solution by modeling the target value as corrupted with asymmetric noise. To learn the regression function, they derive a loss function based on the data model and use gradient descent to find the solution. They use real and synthetic data to demonstrate the proposed approach.


**Summary Of The Review:**

Overall the authors present an interesting problem with a novel solution. They prove that the gradient of the proposed loss is unbiased and consistent. Then, they demonstrate that the method works on synthetic and real data. This is a valuable contribution to the community, and I believe that the paper should be accepted.

---

> ### Author Response · Authors · 2022-11-11
> **Thank you**
>
> Thank you for the helpful comments and suggestions. Please see below for the answers to your questions and comments.
>
> >Page 4 results of figure 2, I assume that the data in the test set is not incomplete, but this is not explained.
>
> The test sets do not have incomplete observations. We have added this description in the Results paragraph in the updated Section 4.1.1.
>
> >P9 the results related to table 2, can be viewed as a classification task, so why not compare to baselines that focus on asymmetric classification?
>
> In this case study, we can not prepare test data that have no incomplete observations. Therefore, we can not evaluate the performance metric for regression well since the value of the metric deteriorates significantly when incomplete observations occur on the test data. From this, the evaluation was done with the classification metric, which a human expert can manually evaluate. Also, note that the real-valued outputs of Actigraph themselves are also meaningful for the task. Thus, it should be regression.
>
> >For clarity, please mention what K is in the caption of figure 2.
>
> >Please expand on the caption of Table 2.
>
> >Why is the word Rate capitalized?
>
> We updated the main text according to your suggestions.

---

> > ### Comment · Reviewer_mxCJ · 2022-11-21
> > **After rebutal**
> >
> > After reading the response and changes, I keep my score unchanged.

---

### Official Review · Reviewer_QdLV · 2022-10-23

**Confidence:** 3
**Correctness:** 2
**Technical Novelty And Significance:** 1
**Empirical Novelty And Significance:** Not applicable
**Recommendation:** 1

**Clarity, Quality, Novelty And Reproducibility:**

The paper is explained in a relatively clear way, but the quality or novelty is not sufficient for publication at ICLR. There are also problematic assumptions / results which I have brought up in the above section.

**Strength And Weaknesses:**

Strength:
The idea is explained well. Their work is motivated by real-world use cases which makes it interesting.

Weakness:
In general I do not really buy this idea for addressing asymmetrically corrupted dataset, and the assumptions they made seem problematic. On a high level, completely ignoring the lower-valued labels, no matter they are true low values or come from incomplete observations, could potentially lose some information. Couple of questions are in order.

1. As the authors showed in the experiments, Huber regression performed badly. But have you tried using Huber regression as an outlier detection step? Namely, identify observations whose (x,y) relationship differs significantly from others, and remove those from your sample set, and then apply Least Squares regression to the rest of normal data points. The logic is that, I do not think we can completely remove lower-valued labels, but instead, we can do some pre-processing to identify abnormal data points. The high-value / low-value scheme does not seem to well supported.

2. Section 2.1 is not needed in my view. It is common knowledge and should be described very briefly.

3. Based on your definition of upper-side labeled data, their noise \epsion_s >=0. But technically \epsilon_s could also be negative. So you are missing a portion of complete observations by restricting to the upper-side labeled data.

4. Lemma 2.2 also seems problematic. Ideally when there is no corruption, namely, (x,y), we want to use all the data, not just data with f(x) <= y. Eq. (5) doesn't seem to establish the consistent estimator as the authors claimed.

5. The authors claimed that for some loss functions, when y<f(x), the gradient of the loss function does not depend on y, e.g., Eq. (8). But notice that even when y>f(x), the gradient of this type of loss also does not depend on y. So it is about the loss function itself, not the upper or lower sides of the labels. When you switch to squared loss (which was used in your Proposal-2 in the experimental section and thus violated your assumption here), the gradient would depend on y no matter you are in the upper or lower sided region.

6. In order to identify the upper or lower sides, you need to do an iterative process where you use the f_t from previous iteration to define upper or lower region. Does this unnecessarily complicate the problem?


**Summary Of The Paper:**

This paper considered a regression problem where incomplete observations have labels biased towards lower values. The authors proposed an algorithm which utilizes only high-valued labels and discarded the lower-valued labels. They conducted several experiments on both synthetic and real-world datasets to demonstrate their algorithm.

**Summary Of The Review:**

Overall I do not buy the idea or arguments made in this paper. For detailed comments please look at the Strength and Weakness section above.

---

> ### Author Response · Authors · 2022-11-11
> **Thank you**
>
> Thank you for the helpful comments and suggestions. Please see below for the answers to your questions and comments.
>
>
> >As the authors showed in the experiments, Huber regression performed badly. But have you tried using Huber regression as an outlier detection step? Namely, identify observations whose (x,y) relationship differs significantly from others, and remove those from your sample set, and then apply Least Squares regression to the rest of normal data points. The logic is that, I do not think we can completely remove lower-valued labels, but instead, we can do some pre-processing to identify abnormal data points. The high-value / low-value scheme does not seem to well supported.
>
> We can equivalently implement a robust method, MAE (least absolute square), with iterative reweighted least squares [1], which can be seen as outlier detection, elimination (reducing their effect), and applying least squares, as you mentioned. As shown in Lemma 3.4 and Proposition 3.2, they are still biased since such approaches assume a symmetric or equal possibility for outliers on the upper and lower sides. In this sense, it would be possible to implement a similar method to the proposed method by using outlier elimination only for lower-side case ($y<f(x)$). Although the such method and the proposed method could work similarly, we showed the specific construction of the proposed method works well theoretically and experimentally in this paper.
>
> [1] Schlossmacher, E. J. "An iterative technique for absolute deviations curve fitting." Journal of the American Statistical Association 68.344 (1973): 857-859.
>
>
> >Based on your definition of upper-side labeled data, their noise $\epsilon_s >=0$. But technically $\epsilon_s$ could also be negative. So you are missing a portion of complete observations by restricting to the upper-side labeled data.
>
> >Lemma 2.2 also seems problematic. Ideally when there is no corruption, namely, (x,y), we want to use all the data, not just data with $f(x) <= y$. Eq. (5) doesn’t seem to establish the consistent estimator as the authors claimed.
>
> Based on some technical assumptions, including $\epsilon_s >=0$ for upper-side labeled data, we derive an algorithm and give a theoretical guarantee for its unbiasedness and consistency. Although these assumptions are not always satisfied in real-world data, our experiments showed that the proposed method works more effectively than the baselines even when the assumptions are not satisfied exactly. The assumption $\epsilon_s >=0$ comes from the condition $f\in\cal{F}'$, which represents our natural expectation that the regression function $f$ well approximates $f^∗$. It is not a strong assumption if we use models with enough expressive power, such as deep neural networks, and enough training data. We also have results with $K=0$ % which means there is no corruption for synthetic data experiments. For LowNoise, HighNoise, and Breathing, the MAEs are 0.57 (Proposed) vs. 0.24 (MSE), 0.79 (Proposed) vs. 0.77 (MSE), 0.45 (Proposed) vs. 0.41 (MAE), respectively. Except for LowNoise task, the proposed method worked comparably to MSE even when $K=0$ %. Since LowNoise with $K=0$ % is very clean and simple data with a linear relationship between $x$ and $y$ and little additive white Gaussian noise, MSE can perform much better than robust methods, including the proposed method. We again note that this paper addresses when there is corruption. If there is no corruption, we can use simple loss functions, such as squared loss.
>
>
> >The authors claimed that for some loss functions, when y<f(x), the gradient of the loss function does not depend on y, e.g., Eq. (8). But notice that even when y>f(x), the gradient of this type of loss also does not depend on y. So it is about the loss function itself, not the upper or lower sides of the labels. When you switch to squared loss (which was used in your Proposal-2 in the experimental section and thus violated your assumption here), the gradient would depend on y no matter you are in the upper or lower sided region.
>
> Condition 3.1 is only applied to the loss function for lower-side labeled data ($y<f(x)$). We can use different functions for the upper and lower sides and can use any function as a loss function for upper-side labeled data ($f(x) \leq y$). We actually used squared loss for upper-side labeled data ($f(x) \leq y$) and absolute loss for lower-side labeled data ($y<f(x)$) in the experiments in Section 4.1.2.

---

### Official Review · Reviewer_gU2i · 2022-10-23

**Confidence:** 3
**Correctness:** 4
**Technical Novelty And Significance:** 3
**Empirical Novelty And Significance:** 3
**Recommendation:** 6

**Clarity, Quality, Novelty And Reproducibility:**

This draft is well-organized and easy to follow. The problem in this draft is novel and interesting. The proposed method seems promising, but the implementation is not discussed in sufficient detail.

**Strength And Weaknesses:**

Strength:
+ The work is well motivated by many sensor-based applications and covers a wide range of real-world applications. To address this task, the proposed method appears promising. Also, empirical results support the proposed method's effectiveness.
+ The proposed method has nice theoretical properties. According to certain assumptions, the proposed approach produces an unbiased estimator with asymmetrically corrupted data and thus obtains a well-generalized model.

Weakness:
- The estimation of $\pi_{up}$ is heuristic and changes according to the training procedure based on the validation set. Therefore, the proposed method and the theoretical results do not end-to-end match exactly. Due to the small validation data set, the estimation can be unreliable and unstable due to overfitting.
- This method can only be applied to absolute loss and pinball loss, which limits its use.

There is a question regarding the implementation of the proposed method. The proposed method relays on an adaptive estimation of $/pi_[up]$ and an upper-side labeled sample set based on the validation set. What is the size of the validation set and how does it affect the performance of the proposed method? How to avoid the overfit to the validation data?

**Summary Of The Paper:**

This draft studies a special weakly supervised regression problem in which the labels are generalized by sensors. Therefore, the low value displayed by the sensor could either be actual or suggest that the label is missing. The author formulates this problem as a regression from asymmetrically corrupted data and proposes a new approach based on an unbiased gradient estimator to solve it. Both synthetic and real-world experiments demonstrate the effectiveness of the proposed approach.

**Summary Of The Review:**

This draft studies a well-motivated problem that covers a wide range of real-world sensor-based applications. The proposed method constructs an unbiased gradient estimator with asymmetrically corrupted data under certain mild assumptions. There is a theoretical basis for the proposed method and empirical studies support its effectiveness.

---

> ### Author Response · Authors · 2022-11-11
> **Thank you**
>
> Thank you for the helpful comments and suggestions. Please see below for the answers to your questions and comments.
>
>
> >The estimation of $\pi_{{\mathrm{up}}}$ is heuristic and changes according to the training procedure based on the validation set. Therefore, the proposed method and the theoretical results do not end-to-end match exactly. Due to the small validation data set, the estimation can be unreliable and unstable due to overfitting.
>
> Our paper provides a grid-search-based approach to estimating the hyperparameter $\pi_{{\mathrm{up}}}$. We showed that it was effective in our experiments even for real-world data. In particular, when the proportion of incomplete training/validation samples is low, this approach is expected to work well. In addition, the experiments showed that the proposed method worked more effectively than the baselines, even when $K = 75$ %, where many samples are incomplete. In PU learning studies, most papers, including the latest ones, rather assume that a hyperparameter corresponding to $\pi_{{\mathrm{up}}}$ is given.
>
>
> >This method can only be applied to absolute loss and pinball loss, which limits its use.
>
> Condition 3.1 is only applied to the loss function for lower-side labeled data ($y<f(x)$). Thus, we can use any function as a loss function for upper-side labeled data ($f(x) \leq y$). We actually used squared loss for upper-side labeled data ($f(x) \leq y$) and absolute loss for lower-side labeled data ($y<f(x)$) in the experiments in Section 4.1.2. Also, note that absolute loss, pinball loss, and other losses that satisfy Condition 3.1 can be generally applied and do not limit the use of the proposed approach, as discussed just below Condition 3.1 in the main text. Rather, they give a clear guide as to what loss function should be used to deal with asymmetrically corrupted data.
>
>
> >There is a question regarding the implementation of the proposed method. The proposed method relays on an adaptive estimation of $\pi_{{\mathrm{up}}}$ and an upper-side labeled sample set based on the validation set. What is the size of the validation set and how does it affect the performance of the proposed method? How to avoid the overfit to the validation data?
>
> We used a randomly sampled $20$ % of the training set as a validation set to choose the best value for $\pi_{{\mathrm{up}}}$, as stated in Section 4.1.2. Also, as discussed above, this approach is robust enough for the high proportion of incomplete training/validation samples even when $K = 75$ %.

---

> ### Author Response · Authors · 2022-11-19
> **Stability of hyperparameter estimation**
>
> To demonstrate the robustness of our validation-set-based approach to estimating the hyperparameter $\pi_{{\mathrm{up}}}$, we have added the performance of the proposed method over different sizes of the validation set in Fig. 4 in Appendix E.3.
> It shows that the proposed method does not degrade its performance much, even when we use only $1$ % of the training set as the validation set. This analysis demonstrates that the proposed approach is robust enough also for the small size of the validation set as well as the high proportion of incomplete validation samples.
> In Fig. 5 in Appendix E.3, we have also added charts similar to Fig. 2 (the error in prediction) when we used $1$ % of the training set as the validation set. We can see that even in this case, the proposed method achieved unbiased learning (the average error shown by the blue solid line is approximately zero.).

---

### Official Review · Reviewer_Sduq · 2022-10-26

**Confidence:** 3
**Correctness:** 3
**Technical Novelty And Significance:** 2
**Empirical Novelty And Significance:** 2
**Recommendation:** 5

**Clarity, Quality, Novelty And Reproducibility:**

Gloablly good, except for the following points
- the way conditional expectation are presented is confusing to me
- in Eq. (9), (10), shouldn't it be $y'$ instead of $y$?
- Lem 3.4: shouldn't $\eta$ be $1/2$ since the noise is symmetric?

**Strength And Weaknesses:**

**Strengths**
- the paper is globally clear and well written
- the problem studied is of interest and the proposed approach is natural

**Weaknesses**
- it seems to me that the interest of the approach is completely shortcut by the choice of loss functions, which do not depend on the label. This is a very strong limitation of the approach to me
- in the same vein, could the authors think about other examples where to apply a similar method? This could make the contribution of greater interest
- the experiments are pretty rudimentary, especially on real data. For instance how would a model with censorship behave, when given the information if the output has been altered or not?
- I am also pointing out the literature on Median-of-Means (MoM)-based methods for robust regression, see in particular [1], that do not assume the outliers to be symmetric and could be interesting to benchmark

[1] Robust classification via MOM minimization, Lecué et al. 2020

**Summary Of The Paper:**

This paper studies the problem of learning a decision function when the output might be corrupted by the fact that the sensor in charge of collecting it has failed to record it properly, underestimating it. Without a debiasing procedure, the decision function naturally underestimate the magnitude of the event. Under the assumption that the loss function does not depend on the prediction (when the latter is greater than the output), the authors propose an estimator of the gradient that is unbiased. Experiments complement the paper.

**Summary Of The Review:**

Overall, I feel the contribution of this paper might not be enough to warrant acceptance. In particular, the restricted choice of loss functions is shortcuting the interest of the approach. The experiments need strengthening.

---

> ### Author Response · Authors · 2022-11-11
> **Thank you**
>
> Thank you for the helpful comments and suggestions. Please see below for the answers to your questions and comments.
>
> >it seems to me that the interest of the approach is completely shortcut by the choice of loss functions, which do not depend on the label. This is a very strong limitation of the approach to me
>
> Loss functions do depend on the label. What does not depend on the label is the gradient of the loss functions. This independency is not a limitation but rather the key advantage of the proposed approach in dealing with asymmetrically-corrupted data. The class of loss functions satisfying Condition 3.1 is broad since Condition 3.1 is only applied to the loss function for lower-side labeled data ($y<f(x)$). Thus, we can use any function as a loss function for upper-side labeled data ($f(x) \leq y$). We actually used squared loss for upper-side labeled data ($f(x) \leq y$) and absolute loss for lower-side labeled data ($y<f(x)$) in the experiments in Section 4.1.2. Also, note that the absolute loss and pinball loss themselves work well on real data, as discussed just below Condition 3.1 in the main text.
>
>
> >in the same vein, could the authors think about other examples where to apply a similar method? This could make the contribution of greater interest
>
> U2 regression has broad application fields in scenarios we generally handle sensors, such as medicine and healthcare, as discussed in the Introduction. Other applications are in Appendix H, such as statistical downscaling. Also, in Appendix C, we show the opposite asymmetric corruption, in which labels for some observations may become inconsistently higher than those for typical observations. This can be handled as learning from lower-side labeled data and unlabeled data, i.e., LU regression. Such a scenario can be found when we use sensor values as explanatory variables, where the label sensor has ideal coverage, but explanatory sensors have smaller coverage. In that case, there are unidentifiable incomplete observations in explanatory variables.
>
>
> >the experiments are pretty rudimentary, especially on real data. For instance how would a model with censorship behave, when given the information if the output has been altered or not?
>
> Whether the output is incomplete cannot be known in real-world data. This is our main claim for motivating the proposed approach, where a low value of the label can either mean that the actual magnitude of the phenomenon has been low or that the sensor has made an incomplete observation, and there are no clues that allow us to tell which is the case.
>
>
> >I am also pointing out the literature on Median-of-Means (MoM)-based methods for robust regression, see in particular [1], that do not assume the outliers to be symmetric and could be interesting to benchmark
> >[1] Robust classification via MOM minimization, Lecué et al. 2020
>
> Median-based approaches, including MoM and MAE, can handle some portion of asymmetric noise. However, in our scenario with incomplete observations, the expected loss has bias even for such approaches, as shown in Lemma 3.4. This is because they assume a symmetric or equal possibility for outliers on the upper and lower sides. The proposed method does not assume the possibility of outliers on the upper side, which makes our solution unbiased and acts as implicit regularization.

---

> > ### Comment · Reviewer_Sduq · 2022-11-22
> > **Post rebuttal**
> >
> > I thank the authors for their feedback. I have read the other reviews and answers and my stance on the paper has not changed.

---

### Decision · Program_Chairs · 2023-01-20

**Decision:**

Reject

**Justification For Why Not Higher Score:**

- More perspective could be given regarding the dependence (or not) on the loss function, the deeper connection with classification with asymmetrical labels
- As per reviewers comments there is work to be done to better position the problem and its motivations

**Justification For Why Not Lower Score:**

- The maths are correct
- the results are promising


**Metareview: Summary, Strengths And Weaknesses:**

This paper introduces a gradient-based learning scenario for regression when the labels/target obtained are corrupted by a negative noise. The authors show that under some appropriate assumptions they are are capable of computing an unbiased stochastic gradient estimator from the corrupted data, that allows them to perform the optimization procedure required to learn a regression model

+ clear paper, easy to follow
+ an optimization method that is proven from a statistical point of view
- not clear how the proposed works departs from what is done in the classification setting with asymmetric noise (see part of the literature cited in the discussion section of the paper): a folk strategy is to iterative learn an unbiased update vector (here a gradient) of some base non-noisy formulation; this is exactly what the authors do here
- as per the reviewers comments, the authors fail to motivate broad applicability of their setting and contributions
- also, it is not clear how dependent on the loss function used is the contribution: what could be the minimal set of assumptions applying to the loss function for a training method to still be working?